

# Mixing and ageing in the polar lower stratosphere in winter 2015/2016

Jens Krause[1], Peter Hoor[1], Andreas Engel[2], Felix Plöger[3], Jens-Uwe Grooß[3], Harald Bönisch[2,4], Timo Keber[2], Björn-Martin Sinnhuber[4], Wolfgang Woiwode[4], and Hermann Oelhaf[4]

[1]Institute for Atmospheric Physics, Johannes Gutenberg-University of Mainz, Germany
[2]Institute for Atmospheric and Environmental Sciences, University of Frankfurt, Germany
[3]Institute of Energy and Climate Research (IEK-7), FZ Jülich, Germany
[4]Institute of Meteorology and Climate Research (IMK), Karlsruhe Institute of Technology (KIT), Germany

*Correspondence to:* Jens Krause (krauseje@uni-mainz.de)

**Abstract.** We present data from winter 2015/2016, which were measured during the POLSTRACC (The Polar Stratosphere in a Changing Climate) aircraft campaign between December 2015 and March 2016. The focus of this work is on the role of transport and mixing between aged and potentially chemically processed air masses from the stratosphere with mid and low latitude air mass fractions with small transit times originating at the tropical lower stratosphere. By combining measurements

of CO, $N_2O$ and $SF_6$ we estimate the evolution of the relative contributions of transport and mixing to the UTLS composition over the course of the winter.

We find an increasing influence of aged stratospheric air partly from the vortex as indicated by decreasing $N_2O$ and $SF_6$ values over the course of winter. Surprisingly we also found a mean increase of CO by $(3.00 \pm 1.64)$ ppb$_V$ from January to March relative to $N_2O$ in the lower stratosphere. We show that this increase of CO is consistent with an increased mixing of

tropospheric air as part of the fast transport mechanism in the lower stratosphere surf zone. The analysed air masses were partly affected by air masses which originated at the tropical tropopause and were quasi-horizontally mixed into higher latitudes.

This increase of the tropospheric air fraction partly compensates for ageing of the UTLS due to the diabatic descent of air masses from the vortex by horizontally mixed, tropospheric influenced air masses. This is consistent with simulated age spectra from the Chemical Lagrangian Model of the Stratosphere (CLaMS), which show a respective fractional increase of tropospheric

air with short transit times lower than six months and a simultaneous increase of aged air from deep stratospheric and vortex regions with transit times larger than two years.

We thus conclude that the lowermost stratosphere in winter 2015/16 was affected by aged air from the deep stratosphere and vortex region. These air masses were significantly affected by increased mixing from the lower latitudes, which led to a simultaneous increase of the fraction of young air in the Arctic lowermost stratosphere over the course of winter.

## 1 Introduction

The upper troposphere / lower stratosphere (UTLS) (Fig. 1) region at high latitudes during winter is strongly affected by the evolution of the polar vortex. Diabatic descent, which is most pronounced inside the polar vortex adds to the diabatic down-





welling of the Brewer-Dobson circulation (Brewer, 1949; Dobson, 1956) in mid and high latitudes as response to the breaking of planetary and gravity waves (Haynes et al., 1991; Plumb, 2002; Butchart, 2014) in the upper stratosphere and mesosphere, respectively. This downwelling leads to an increasing contribution of deep stratospheric air masses in the UTLS over the course of winter where they contribute to the composition of the lowermost stratosphere (LMS) (Rosenfield et al., 1994; Holton et al., 1995).

Chemically these air masses differ from the composition of the LMS, since they are potentially affected by ozone depleting catalytic cycles (Solomon, 1999). Since the air inside the polar vortex is largely isolated and exhibits a strong diabatic descent due to radiative cooling and the wave-driven Brewer-Dobson circulation this leads to an increased fraction of air masses with a high mean age of air in the UTLS of high latitudes (e.g. Engel et al., 2002; Ploeger et al., 2015).

The mean age of air is defined as the first moment of the transit time distribution (or the so-called age spectrum) (Hall and Plumb, 1994; Waugh et al., 1997). Mean age can be determined from the observation of long-lived tracers, which ideally have no sources or sinks in the stratosphere and of which the temporal evolution of the mixing ratio at the tropical tropopause is well known (Waugh et al., 1997). Notably, the mean age is a bad descriptor for the full age spectrum, which is highly skewed (e.g. Hall and Plumb, 1994) and sometimes even multimodal (Andrews et al., 1999; Boenisch et al., 2009). For the estimate of the potential chemical impact of species particularly with lifetimes on the order of weeks to only a few months the mean age is insufficient and the full spectrum is needed (Schoeberl et al., 2000), which is however only available under very idealized conditions (Schoeberl et al., 2005; Ehhalt et al., 2007).

Observations of $SF_6$, $N_2O$ and $CO_2$ from the ER-2 aircraft show that the mean age at northern high latitudes at an altitude of 20 km is on the order of 4-6 years (Andrews et al., 2001; Engel et al., 2002). Satellite observations of $SF_6$ confirm this and show further a strong interannual variability of the mean age in northern high latitudes (Stiller et al., 2008, 2012; Haenel et al., 2015). The observations also indicate a potential transport of mesospheric air to lower altitudes (Engel et al., 2006b; Ray et al., 2017), which however strongly depends on the strength and persistence of the Arctic polar vortex during the individual winters. In addition to diabatic descent inside and outside the polar vortex quasi-isentropic mixing from lower latitudes leads to a contribution of relatively young air to the UTLS. As a result a seasonal cycle of the chemical composition of the UTLS up to $\Theta = 430$ K establishes with a more tropospheric character during northern summer / autumn and a more stratospheric characteristic in late winter/spring (Hegglin and Shepherd, 2007). The chemical composition and age structure of the extratropical UTLS (ExUTLS) are affected by the competing diabatic downwelling of aged air and rapid quasi isentropic mixing down to the tropopause (Hoor et al., 2005; Engel et al., 2006a; Boenisch et al., 2009; Garny et al., 2014). The region between $\Theta = 380$ K and the bottom of the subtropical pipe around $\Theta = 450$ K (Palazzi et al., 2011) is a key region for the transition between these transport regimes. The $\Theta = 380$ K isentrope coincides with the tropical tropopause and is therefore directly affected by diabatic vertical transport of tropospheric air through the tropical transition layer (TTL) (Fueglistaler et al., 2009) into the stratosphere. These air masses are rapidly mixed quasi horizontally by breaking planetary waves with air from high latitudes in addition to the shallow branch of the Brewer-Dobson circulation (Birner and Bönisch, 2011; Abalos et al., 2013). This rapid transport modifies the abundance of particularly water vapour and ozone in this region (Rosenlof et al., 1997; Randel et al., 2006; Hegglin and Shepherd, 2007; Pan et al., 2007).





To estimate the radiative effects of these species with large gradients at the tropopause, the details of mixing are essential (Forster and Shine, 1997, 2002; Riese et al., 2012). Uncertainties arising only from uncertainties in mixing may lead to significant uncertainties of the radiative forcing, which are on the order of 0.5 W m$^{-2}$ (Riese et al., 2012).

In our study we focus on the transition of the tracer composition in the subvortex region up to $\Theta = 410$K during winter 2015/2016. We will quantify the effects of quasi-isentropic mixing from the tropics and diabatic downwelling and its effect on the chemical composition as well as the evolution of the age spectrum and the mean age in this region.

## 2 Meteorological conditions during winter 2015/16

The early Arctic winter 2015/16 was the coldest winter in the lower stratosphere (LS) since 1948. These extreme cold conditions could establish due to a strong and cold Arctic polar vortex which developed in November 2015 due to very low planetary wave activity in the stratosphere (Matthias et al., 2016). From late December 2015 to early February 2016 the temperatures at $\Theta = 490$ K decreased below 189 K. Therefore strong dehydration and denitrification was seen in low $H_2O$ and $HNO_3$ volume mixing ratios, which finally led to a strong chlorine activation in early winter. Using MLS data the chemical influence of the vortex could be observed to isentropes below $\Theta = 400$ K (Manney and Lawrence, 2016).

The major final warming (MFW) occurred on 5 March 2016 which led to a split of the vortex one week later. This early final warming was unusual, as only five other MFWs since 1958 appeared before middle of March. Due to this early warming air masses in the polar lower stratosphere were mixed with non-vortex air and prevented chemical ozone depletion reaching record low values during winter 2015/16 (Manney and Lawrence, 2016, and references therein).

The winter 2015/16 was characterised by an unprecedented anomaly of the quasi biannual oscillation (QBO) with a westward jet formed within the eastward phase in the lower stratosphere (Newman et al., 2016; Osprey et al., 2016). Since the QBO affects the zonal wind direction in the tropical lower stratosphere (Niwano et al., 2003) its strength and phase is crucial for stratospheric transport processes and westerly phases are related to strong and cold polar vortices (Holton and Tan, 1980).

Further the winter 2015/16 was also affected by a strong warm phase of the El-Niño Southern Oscillation (ENSO) (McPhaden et al., 2015). A direct influence on the polar vortex is still under debate, but according to Matthias et al. (2016) this strong El-Niño is suggested to account for a weakening of the polar vortex.

## 3 Project overview and measurements

This work will address the evolution of composition, age structure and the influence of transport and mixing of air masses in the lower stratosphere. The composition of air masses inside the LS, which is affected by diabatic descent of upper stratospheric air masses, irreversibly mixed with younger air from the TTL is analysed by combining measurements of in-situ data with model calculations of the Chemical Lagrangian Model of the Stratosphere (CLaMS) (McKenna et al., 2002; Grooß et al., 2014; Ploeger et al., 2015).





## 3.1 The POLSTRACC campaign 2015/16

The data presented in this study were obtained during the POLSTRACC (Polar Stratosphere in a Changing Climate) mission, which was part of the combined PGS (POLSTRACC/GW-LCYCLE/SALSA) framework. The main objectives of the POL-STRACC mission were the investigation of structure, composition and dynamics of the Arctic LMS and processes involving

chemical ozone depletion and polar stratospheric clouds in the Arctic winter UTLS. In total 17 scientific flights were performed from December 2015 until end of March 2016 on board the new German research aircraft HALO (High Altitude Long Range) from Oberpfaffenhofen, Germany (48.05 °N, 11.16 °E) and Kiruna, Sweden (67.49 °N, 20.19 °E) covering the region from 25 °N to 87 °N and 24 °E to 80 °W (Fig. 2). Typical flight altitudes ranged from 10 km asl[1] to 14.5 km asl corresponding to potential temperatures in the stratosphere from $\Theta = 320$ K up to $\Theta = 410$ K. The total flight time was about 157 hours,

of which 19 hours were spent in December 2015, 62 hours were spent from January to February and 76 from February to March, respectively. For this study we focus on Arctic measurements starting from Kiruna, which took place during two campaign phases, representing flights from 12. January 2016 to 02. February 2016 (phase 1) and from 26. February 2016 to the 18. March 2016 (phase 2). For the aim of this work we use approximately 50 hours of measurements of those flights which were conducted to probe air masses predominantly underneath the polar vortex above PV = 7 PVU.

The research aircraft HALO is a modified business jet type Gulfstream G-550. It has a maximum range of 12500 km with a maximum altitude of 15.5 km and can carry about up to 3 tons of scientific payload. The payload was a combination of different remote sensing (e.g. WALES lidar (Wirth et al., 2009; Fix et al., 2016), Väisälä RD 49 dropsondes and GLORIA limb sounder (Friedl-Vallon et al., 2014; Kaufmann et al., 2015)) and in-situ instruments of trace gases with different lifetimes, sources and sinks.

## 3.2 In-situ trace gas measurements

In this study we analyse measurements of $N_2O$, CO, which were measured with the TRIHOP instrument (Müller et al., 2015) and $SF_6$ by the GhOST-MS instrument (Sala et al., 2014). For our analysis the data are synchronised to a common time resolution of 10 seconds or 0.1 Hz respectively, corresponding to a horizontal resolution of 2.5 km at typical HALO flight speeds.

GhOST data is available with a resolution of 60 seconds at an integration time of one second which leads to a horizontal resolution of 15 km.

### 3.2.1 The TRIHOP instrument

The TRIHOP instrument (Schiller et al., 2008) is an infra red absorption laser spectrometer with three quantum cascade lasers

(QCL) operating between wavenumbers 1269 $cm^{-1}$ and 2184 $cm^{-1}$. It was set up to measure CO, $N_2O$ and $CH_4$ during the POLSTRACC campaign. To quantify mixing ratios in the order of $ppb_V$ the instrument uses a multi pass White-cell which

---

[1]above sea level

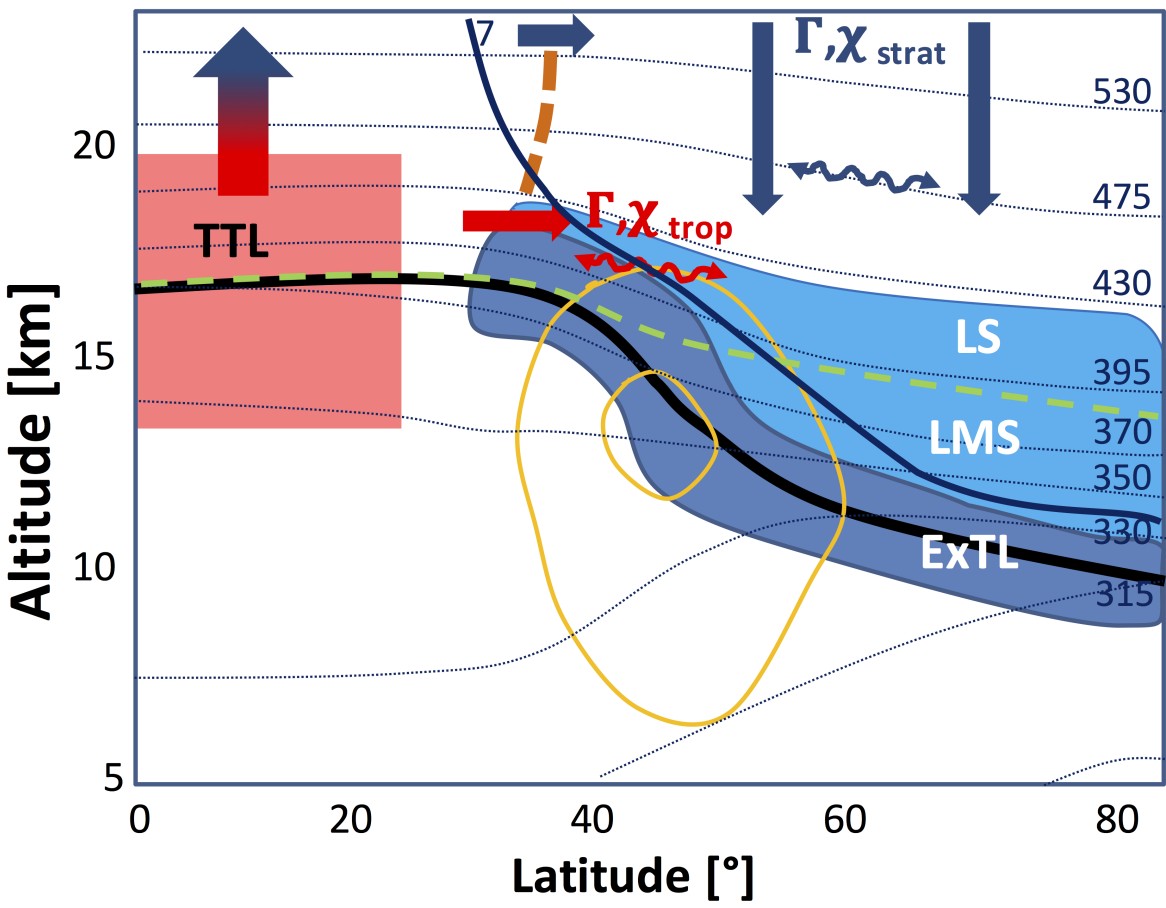

**Figure 1.** Cross section of the northern hemispheric UTLS (Upper Troposphere / Lower Stratosphere) region, adapted from Riese et al. (2014) and Müller et al. (2016). The thermal tropopause is denoted by the thick black solid line. The measurement region is depicted as blue box subdivided into the extratropical tropopause layer (ExTL), the lowermost stratosphere (LMS) and the lower stratosphere (LS). LMS and LS are separated by the 380 K isentrope (green dashed line). Transport pathways of air masses are denoted by coloured, thick arrows from the tropical tropopause layer (TTL) (red) and the polar Arctic upper stratosphere (blue) with respective mean age $\Gamma$ and trace gas volume mixing ratio $\chi$. Quasi horizontal mixing is represented by wavy double side arrows, indicating no net mass transport of air masses. Dotted lines are isentropes in K, the solid dark blue line indicates the 7 PVU contour, which is used to separate the regime of the ExTL from the LMS and LS (for details see text). Thin orange contour lines depict the zonal view of the jet stream.





**Figure 2.** Flight tracks during the POLSTRACC campaign. Blue colours indicate flights during phase 1 (12. January-02. February), red colours indicate flights during phase 2 (26. February-18. March). Only flights are shown, which were used for the analysis. For details see Sect. 3.1.



is pressure regulated at a pressure of 30 mbar to minimize pressure broadening of the absorption lines. The measurements were performed with an integration time of 1.5 seconds per species. The three species are subsequently measured during a full cycle which finally leads to a time resolution of 7 seconds due to additional latency times when the channels are switched. The instrument is regularly calibrated in-flight against compressed standards of ambient air which are in turn calibrated prior and

after the campaign against primary standards, connected to the World Meteorological Organisation Global Atmosphere Watch Central Calibration Laboratory (WMO GAW CCL) scale (X2007) for greenhouse gases. During POLSTRACC it was possible to achieve a $(2\sigma)$ precision of CO, $N_2O$ and $CH_4$ of 1.15, 1.84 and 9.46 $ppb_V$ respectively.

### 3.2.2   GhOST-MS in-situ measurements

The GHOST-MS instrument is a two channel gas chromatograph for airborne measurements of trace gases. One channel uses

a mass spectrometer (Agilent MSD 5975) for the detection of atmospheric trace gases at a time resolution of four minutes. This channel uses negative ion chemical ionization as described in Sala et al. (2014) to measure brominated hydrocarbons. The other channel measures $SF_6$ and CFC-12 using an ECD (Electron capture detector) with a time resolution of one minute. For the POLSTRACC campaign the precision for $SF_6$ was 0.6% and the precision for CFC-12 was 0.2%.

Mean age of air is inferred from $SF_6$ measurements (Engel et al., 2009). Due to its much higher atmospheric mixing ratio,

the precision of CFC-12 measurements is better than that of $SF_6$ measurements. Prior to calculating mean age, the $SF_6$ time series has therefore been smoothed using the CFC-12, by applying a local (ten minutes of data before and after the time of measurement) fit between CFC-12 and $SF_6$. This procedure removes parts of the instrumental scatter but retains the local information and does not introduce any offset to the mean age values. Mean age derived in this way has an overall precision of better than 0.3 years and an estimated accuracy of 0.6 years, as explained in Engel et al. (2006a). Both $SF_6$ and CFC-12 are

reported on the SIO-2005 scale.

### 3.3   The Chemical Lagrangian Model of the Stratosphere (CLaMS)

The analysis of trace gas measurements is complemented by simulations with the Chemical Lagrangian Model of the Stratosphere CLaMS (McKenna et al., 2002; Konopka et al., 2004). CLaMS is a Lagrangian chemistry transport model, based on forward trajectory calculations and a parameterization of small-scale atmospheric mixing which depends on the deformation

rate of the large-scale flow. The model simulation is driven with meteorological data (e.g., horizontal wind fields) from European Center of Medium Range Weather Forecasts (ECMWF) ERA-Interim reanalysis (Dee et al., 2011) and covers the period 1979-2017. The model uses an isentropic vertical coordinate throughout the stratosphere and the vertical velocity is deduced from the reanalysis total diabatic heating rate. Further details about the model set-up and the included chemical reactions (relevant species here are CO and $N_2O$) are given in Pommrich et al. (2014). This long-term CLaMS simulation has been shown

to reliably represent transport processes in the lower stratosphere for the relevant trace gas species CO and $N_2O$ (Pommrich et al., 2014) as well as for mean age of air (Ploeger et al., 2015).

Recently, a method to calculate the age of air spectrum has been implemented in CLaMS (Ploeger and Birner, 2016), which will be used in the following analysis. The age spectrum is the transit time distribution of air masses for transport from a





control surface (usually take as the tropical tropopause or the Earth's surface) to a given location in the stratosphere (e.g. Hall and Plumb, 1994; Waugh, 2002) and can be related to the Greens function of the transport equation. The calculation method in CLaMS is based on inert tracer pulses, with different tracers released every other month at the surface in the tropics. This method allows calculating time dependent age spectra for the non-stationary atmospheric flow at any location and time in the

model domain (see Ploeger and Birner (2016) for further details).

For the aim of this work also a simulation with full stratospheric chemistry was performed by CLaMS with the setup as described by Grooß et al. (2014). This setup is typically used for periods up to six months. The upper boundary is set to $\Theta = 900$ K potential temperature, where tracers like $O_3$, $N_2O$ and $CO$ are constrained by MLS satellite observations. Due to its Lagrangian formation, a box-trajectory model setup is also possible in which the identical chemistry scheme is used along single air mass

trajectories. This setup is also used here to diagnose chemical pathways and chemical conversion rates. This boxmodel setup is also used here to estimate CO production and loss rates.

## 4   Results

As shown in Hoor et al. (2010) rapid and frequent mixing with tropospheric air mainly affects the region of PV < 7 PVU. To exclude mixing with air masses of recent tropospheric origin or from the exTL (extratropical tropopause layer) we only

selected data above this level of potential vorticity. Therefore the composition of analysed data is mainly affected by isentropically, irreversibly mixed air mass signatures originating out of the tropics and diabatically descended air masses from the upper stratosphere in the polar region. In this analysis we further excluded flights, which were dedicated to the observation of gravity waves.

### 4.1   Tracer distributions and mean age

Figures 3, 4 and 5 show tracer distributions respective as a function of equivalent latitude and potential temperature $\theta$ (Strahan et al., 1999; Hoor et al., 2004; Hegglin et al., 2006). Equivalent latitude is directly linked to the potential vorticity, which is conserved under adiabatic processes (Holton, 2004). Therefore, these coordinates are suitable to account for reversible adiabatic tracer transport.

### 4.1.1   Age of air

An air parcel in the stratosphere is a mixture of fractions of air with different histories, transport pathways and individual transit times. The several transport pathways constitute to an age spectrum or transit time distribution, respectively. The age spectrum can be obtained by calculation of the Green's function of the tracer continuity equation for a conserved and passive species (Hall and Plumb, 1994).

The mean age is defined as the first moment of the transit time distribution. To determine the mean age from measurements, long-lived tracers can be used, which have a well known source distribution at the tropopause and a well defined vertical





gradient in the stratosphere (Hall and Waugh, 1997). Since $SF_6$ is a long-lived inert trace gas with a well known increase of its mean surface mixing ratio, it is a commonly used species for calculations of mean age (Boenisch et al., 2009). The sink of $SF_6$ is in the mesosphere, where it is destroyed by shortwave UV radiation. The lifetime of $SF_6$ is assumed to be 3200 years, but recent studies indicate a significantly shorter lifetime of about 850 years (Ray et al., 2017). This implies that mean age derived

from $SF_6$ may be too old. Especially for polar vortex air, this has been modelled and observed to cause a high bias of up to one year (Ray et al., 2017) or even more in mesospheric air (Engel et al., 2006b). While this may cause a significant offset in mean age for polar vortex air, it is estimated that relative changes in mean age as discussed in this paper can be reliably derived from $SF_6$ observations.

Figure 3 a) and b) show the distribution of mean age calculated from $SF_6$ measurements for phase 1 (January) and phase 2

(February / March), respectively. As is evident on panel (a) during phase 1 the LS is dominated by air masses of mean ages between 0.5 years to less than three years at maximum. The oldest air masses with mean ages older than two years were encountered at largest distances from the tropopause and potential temperatures ranging from $\Theta = 360$ K to $\Theta = 380$ K. In contrast, during phase 2 (panel (b)) in general much older air masses up to five years were found at potential temperatures of $\Theta = 410$ K. These higher potential temperatures at flight altitude are the result of the diabatic descent over the course of winter

and indicate an increasing influence of air masses originating deeper in the stratosphere or from the Arctic polar vortex. To directly compare the temporal evolution of the age of air in the lower stratosphere panel (c) shows the difference of age of air between both phases. It can be seen that the bulk of air inside the LS is getting older between $\Theta = 330$ K and $\Theta = 380$ K. The mean increase is 0.29 years, indicating diabatic downwelling due to the evolution of the polar vortex and thus an increased mean age in late winter.

### 4.1.2 Nitrous oxide

Nitrous oxide ($N_2O$) is a very stable molecule with a lifetime of 123 years (Ko et al., 2013). Its sources are at the surface due to natural and anthropogenic emissions with a very small seasonal variability (Dils et al., 2006). As a result of the well-mixed troposphere and the absence of tropospheric sinks (Ciais et al., 2013) $N_2O$ has a distinct background value in the troposphere, so mixing ratios below this value can be identified as stratospheric influenced (Müller et al., 2015). The tropospheric back-

ground value of $N_2O$ between November 2015 and March 2016 in the northern hemisphere was 329.3 $ppb_V$, measured by the NOAA Global Monitoring Division (NOAA). Its annual increase was found to be 0.78 $ppb_V$ in the last years (Hartmann et al., 2013) with an variability of 3-5 $ppb_V$ (Kort et al., 2011). The main sink reactions of $N_2O$ are due to photolysis in the UV-band (190 nm $\leq \lambda \leq$ 220 nm) and the reaction with $O(^1D)$ which only occurs within the upper stratosphere (Ko et al., 2013). Since there are no sources of $N_2O$ in the stratosphere its profile above the tropopause changes and shows a weak negative vertical

gradient. During winter and spring a stronger negative vertical gradient establishes because of the enhanced diabatic downwelling due to the Brewer-Dobson circulation.

Figure 4 shows the tracer distribution of $N_2O$. During phase 1 (panel(a)) values between 325 $ppb_V$ and 276 $ppb_V$ $N_2O$ were obtained. Consistent with the distribution of mean age lowest values of $N_2O$ less than 200 $ppb_V$ were measured by the end of February above $\Theta = 400$ K (panel (b)). Corresponding to Figure 3 panel (c) shows the difference between phase 2 and phase 1





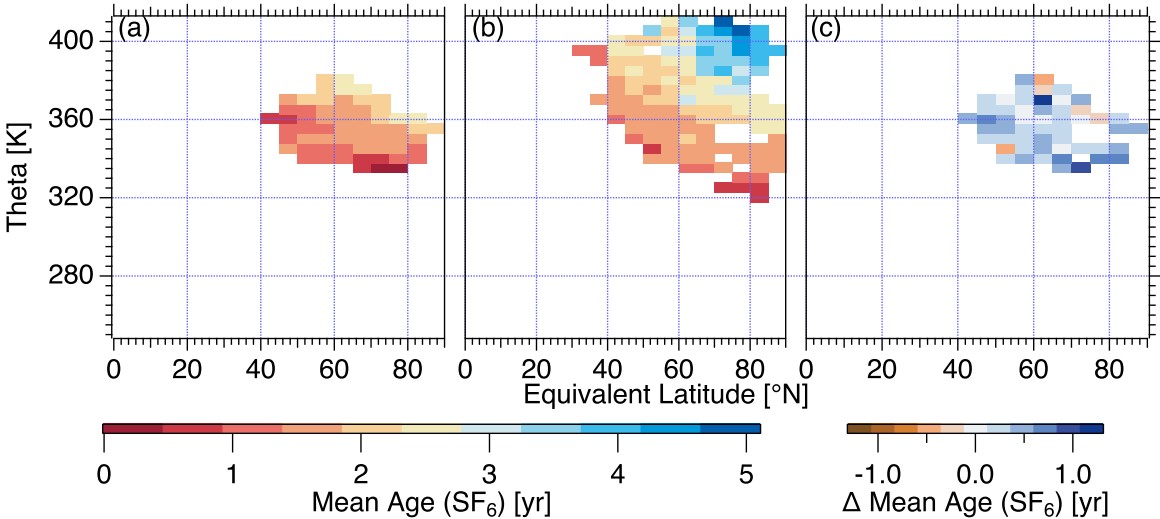

**Figure 3.** Distributions of mean age from $SF_6$ measurements in potential temperature - equivalent latitude coordinates for PV > 7 PVU. Panel (a) shows data for phase 1, (b) for phase 2 and panel (c) shows their absolute difference (phase 2-phase 1). The colour code represents the mean age. Blue colours in panel (c) indicate an increase of mean age in the subvortex region from January to March. Only bins with more than ten data points are shown.

and it is evident that there is a general decrease of $N_2O$ observed in the whole LS, consistent with the measurements of mean age, indicating an enhancement of the diabatic downwelling over the course of winter.

### 4.1.3 Carbon monoxide

Carbon monoxide (CO) is released to the atmosphere mainly through incomplete combustion processes and methane oxidation as the only significant in-situ source. It has therefore a large variability in the troposphere which is also affected by anthropogenic emissions. Due to the high variability of surface emissions CO has variable mixing ratios in the range of 70 $ppb_V$ to 200 $ppb_V$ (Prinn et al., 2000) in the northern hemispheric troposphere with lifetimes on the order of weeks. In the lower stratosphere the main source of CO is the methane oxidation with the OH radical. The main sink reaction is the oxidation with the OH radical where CO gets oxidized into $CO_2$, which leads to a longer lifetime in the order of several months under dark vortex conditions:

$$CO + OH \longrightarrow CO_2 + H \tag{1}$$

We found an equilibrium value of 10-15 $ppb_V$ in winter 2015/2016, depending on the integrated temperature history of the respective air mass in agreement with previous studies (Müller et al., 2016; Herman et al., 1999).

A potential additional source of CO in the Arctic winter stratosphere is the reaction of $CH_4$ with reactive chlorine (Cl) which





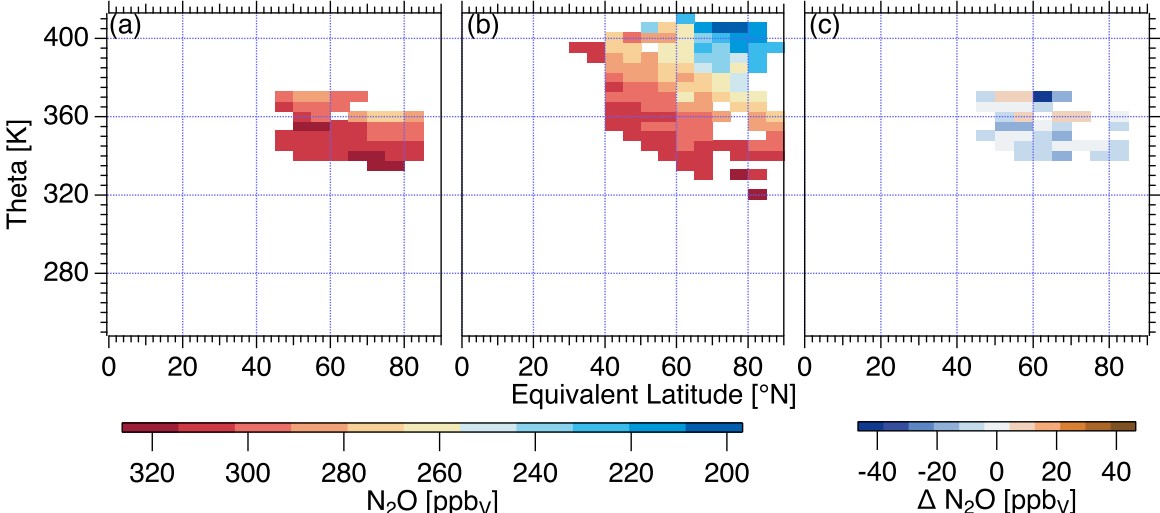

**Figure 4.** As Fig. 3 but for N$_2$O. Negative (blue) values in panel (c) indicate an overall decrease of N$_2$O in the measurement region in accordance with increasing mean age (Fig. 3c).

is nearly insignificant for the lower stratosphere (Flocke et al., 1999). Transport from the mesosphere, where CO is produced from the photolysis of CO$_2$, also provides a potential source of CO via strong diabatic descent during winter under persistent polar vortices (Engel et al., 2006a). These potential influences are discussed in chapter 6.

Figure 5 shows the distribution of CO. During phase 2 (panel (b)) the lowest mixing ratios of 15 ppb$_V$ were found at potential temperatures between $\Theta = 380$ K and $\Theta = 410$ K and equivalent latitudes > 60 °N. As can be seen by the vertical branch of the CO-N$_2$O correlation (Fig. 7), this value is the stratospheric equilibrium during late winter. Phase 1 (panel (a)) values ranged between 60 ppb$_V$ and 17 ppb$_V$, hence the stratospheric background value was not measured in January 2016. A strong tropospheric influence is evident below $\Theta = 340$ K with CO values up to 57 ppb$_V$ at phase 1 and 47 ppb$_V$ at phase 2. Hence the overall distribution of carbon monoxide in the UTLS seems to be consistent to N$_2$O and mean age obtained from SF$_6$ measurements, despite its much shorter lifetime compared to the other species.

However, when comparing the differences of the respective phases (panel c), we see a different behaviour compared to N$_2$O and SF$_6$. We encountered an increase of carbon monoxide mixing ratios over the course of winter, which is at first glance in contradiction to the distributions of mean age and N$_2$O. While the distributions of long-lived tracers SF$_6$ and N$_2$O indicate an ageing of air masses, the increase of short-lived CO indicates a source of CO either from the troposphere or the stratosphere. Note that the increase is observed above $\Theta = 360$ K, whereas below this level a decrease occurs. We will analyse the potential sources of CO in the following and rise the hypothesis that CO increases due to an enhancement of mixing of tropospheric air from the tropical lower stratosphere over the course of winter without a direct strengthening of tropospheric source emissions and mixing ratios from below.



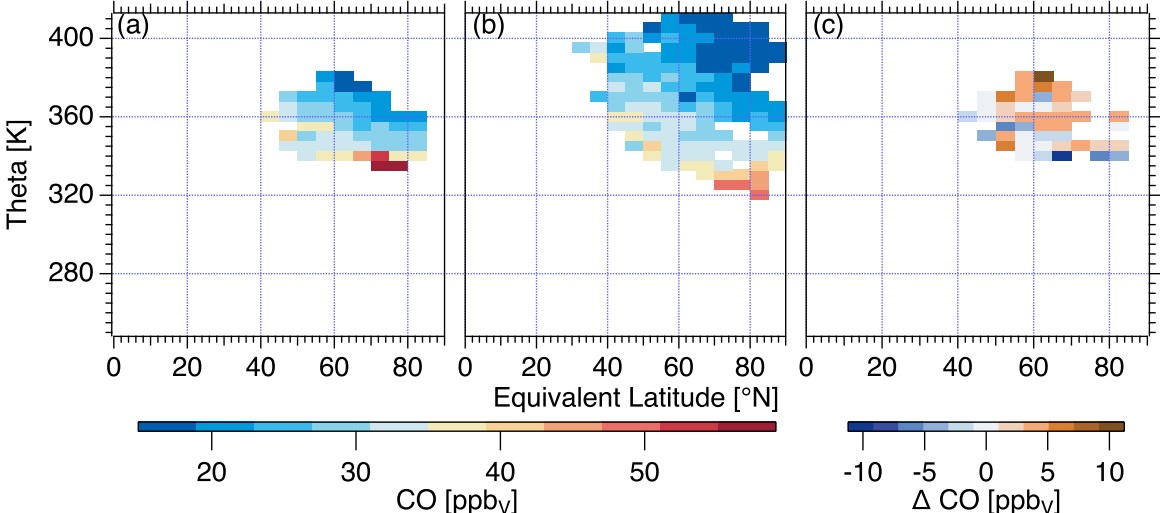

**Figure 5.** As Fig. 3 and Fig. 4 but for CO. Note the positive difference of CO in panel (c) indicating an increase of CO in the measurement region.

## 5 Analysis

We found a decrease of the long lived species $SF_6$ and $N_2O$ with their lowest values in the furthest regions from the troposphere in late winter, which fits well in the general picture of the Brewer-Dobson circulation and the enhanced downwelling in late winter/spring. The contradicting, simultaneous increase of the short lived CO over the course of winter could indicate a strengthening of tropospheric transport by enhanced mixing with fraction of air with low transit times into the lower stratosphere.

### 5.1 Identification of mixing on the basis of tracer-tracer correlations

In the following we will discuss this hypothesis and also other potential sources for the additional CO mixing ratios. We will show that, despite of different potential source regions as the mesosphere or chemical in-situ production, this increase is originating from an enhanced isentropic mixing out of the TTL, interacting with the diabatic descent in the polar stratosphere. To identify mixing processes across the tropopause CO-$O_3$ correlations have been widely used (Fischer et al., 2000; Zahn et al., 2000; Hoor et al., 2002; Pan et al., 2004; Müller et al., 2016). Since ozone is affected by chemical processes particularly in the vortex region we use $N_2O$ as stratospheric tracer instead of ozone. Carbon monoxide, here used as tropospheric tracer also has sources in the mesosphere and via chlorine chemistry in the stratosphere. In the LS the influence of chlorine is small, compared to the reaction with the hydroxyl radical, therefore we investigated the influence of chlorine chemistry regarding methane which leads to the formation of CO. This influence will be discussed in detail later.

To analyse the effect of transport and mixing on the evolution of the UTLS composition we used the $N_2O$-CO relation as shown





in Fig. 6. Tropospheric data are represented by high $N_2O$ (> 328 ppb$_V$) accompanied by high CO values. Thus, stratospheric data appear for $N_2O$ < 328 ppb$_V$. Due to the tropospheric background value of $N_2O$ and the stratospheric equilibrium of CO, the troposphere can be identified as horizontal (high amount of $N_2O$, variable CO) branch and the stratosphere, free of tropospheric influence can be identified as vertical branch (low amount of CO, variable $N_2O$) of the correlation. Without any recent mixing,

the tracer-tracer correlation of $N_2O$ and CO would form an L-shape structure (Fischer et al., 2000). In the presence of actual rapid mixing a straight mixing line between two end members of the correlation establishes (Hoor et al., 2002; Müller et al., 2016). As stratospheric CO will relax towards its stratospheric equilibrium value while $N_2O$ is chemical inert, the initial linear correlation will become curved with time in case of inefficient mixing when the chemical lifetime is shorter than the time scale of mixing. Depending on the strength of mixing relative to the chemical CO sink the curvature will change and is less

pronounced as the mixing gets more efficient. It is important to note that the change of CO relative to a given $N_2O$ value can only be explained by a change of the ratio between mixing and chemical time scales. Mixing alone acts on both tracers $N_2O$ and CO. Therefore a change of the shape of the curve is a direct result of the increased mixing relative to the chemical timescale, which is less efficient when mixing becomes stronger. Panel (d) of Fig. 6 shows additionally the correlation under mesospheric influenced conditions. In this case the correlation would rise to higher CO mixing ratios and lower $N_2O$ mixing

ratios, since $N_2O$ gets destroyed and CO produced in the mesosphere.

Figure 7 shows the $N_2O$-CO correlation for POLSTRACC separated for phase 1 and phase 2, respectively binned in intervals of 5 ppb$_V$ $N_2O$. It is evident that mixing of tropospheric and stratospheric air masses occurs in both phases. During phase 1 (blue curve) CO ranges between 20 ppb$_V$ and 60 ppb$_V$ at $N_2O$ values between 323 ppb$_V$ and 270 ppb$_V$. Notably the red curve (phase 2) shows a steeper gradient with CO values between 43 ppb$_V$ and 15 ppb$_V$ at $N_2O$ values between 323 ppb$_V$ and 180 ppb$_V$.

There are higher CO mixing ratios for $N_2O$ values lower than 310 ppb$_V$ in the later phase 2 of the measurements. Additionally the red curve tends towards an CO equilibrium value of 15.67 ppb$_V$ for $N_2O$ values in the range of 220 ppb$_V$ to 180 ppb$_V$. Most importantly, there is an increase of CO on $N_2O$ isopleths between 313 ppb$_V$ and 273 ppb$_V$ $N_2O$ over the course of winter. This is a remarkable result since we expect that due to the ageing of air inside the lower stratosphere in winter, the CO mixing ratio decreases with time. It is important to note that the correlation along the mixing line which connects tropospheric values

with the stratosphere shows higher CO relative to $N_2O$. Furthermore phase 1 shows higher CO values relative to $N_2O$ compared to phase 2 for $N_2O$ values larger than 313 ppb$_V$. Therefore we can conclude that regarding to the CO-$N_2O$ correlation the direct tropospheric impact was greater in phase 1 than in phase 2, indicating enhanced mixing with tropospheric influenced air originating in the TTL region during phase 2.

A potential mesospheric impact is highly unlikely due to the fact that during phase 2 the $N_2O$-CO correlation tends towards

the equilibrium value in the region of lower $N_2O$ values. This influence will be discussed later in detail.

As shown before the analysed measurement region, which is covered in both phases, lies between potential temperatures of $\Theta = 340$ K and $\Theta = 380$ K. Additionally, the measurement data is filtered for potential vorticity values larger than PV = 7 PVU. Therefore we assume the TTL (Fueglistaler et al., 2009) region (Fig. 1), where most of the tropospheric air masses are trans-

ported into the stratosphere (Schoeberl et al., 2006, and references therein) as main source for the enhanced CO values (Fig.





**Figure 6.** Sketch of tracer-tracer correlations with different lifetimes. Note that the N$_2$O axis is reversed. Panel (a) shows the L-shape structure with an air parcel (red box) on a straight mixing line for fast mixing timescales. The horizontal red line represents the tropospheric N$_2$O background, the blue vertical line the stratospheric CO equilibrium. Panel (b) shows the resulting curve in case of inefficient mixing compared to the chemical lifetime. Panel (c) shows the change of curvature, depending on the strength of mixing and panel (d) shows the influence of the mesosphere on the correlation.





**Figure 7.** $N_2O$-CO correlation for POLSTRACC flights with PV > 7 PVU. The blue curve represents phase 1, the red curve represents phase 2. Data are binned in steps of 5 ppb$_V$ $N_2O$. The variability in each bin is given by the vertical and horizontal lines, respectively.



5 panel (c)). Further on, rapid eddy mixing of air from the TTL leads to an increase of tropospheric tracer signatures in the Arctic region (Rosenlof et al., 1997).

To quantify the increasing influence from tropospheric air masses in the lower stratosphere, we applied a simple mass balance approach to quantify the composition of the lower stratosphere. Therefore, we assume an air parcel in the lower stratosphere

may consist of either upper stratospheric or tropospheric origin (Fig. 1). This mass balance system is solved to get the amount of tropospheric fraction $f_{trop}$ of the measured air.

For a mixing ratio $\chi$ on a specific isentrope $\theta$ we assume

$$\chi(\theta) = f_{\text{trop}} \cdot \chi_{\text{trop}} + f_{\text{strat}} \cdot \chi_{\text{strat}} \tag{2}$$

and

$$f_{\text{trop}} + f_{\text{strat}} = 1 \tag{3}$$

which leads to the tropospheric fraction $f_{\text{trop}}$ based on CO measurements

$$f_{\text{trop}} = \frac{\chi_{\text{CO,m}} - \chi_{\text{CO,strat}}}{\chi_{\text{CO,trop}} - \chi_{\text{CO,strat}}} \tag{4}$$

with $\chi_{\text{CO,m}}$ the measured CO mixing ratio, $\chi_{\text{CO,strat}}$ the stratospheric CO background which was set to 15.7 ppb$_{\text{V}}$ as mean of the vertical branch of the CO-N$_2$O correlation and $\chi_{\text{CO,trop}}$ the tropospheric CO entry value in the TTL.

Earlier studies have shown that CO mixing ratios above the tropical tropopause are at levels between 50 and 60 ppb$_{\text{V}}$ (Herman et al., 1999; Marcy et al., 2007).

The difference of the calculated tropospheric fraction $f_{trop}$ between phase 2 and phase 1 is shown in Fig. 8 as a function of N$_2$O, which acts as a quasi vertical coordinate. The CO increase over the course of winter corresponds to an increase by $f_{trop}$ of 6.8(3.7)% between 313 ppb$_{\text{V}}$ and 273 ppb$_{\text{V}}$ N$_2$O. Note that additionally the tropospheric fraction decreases towards

more tropospheric N$_2$O values from phase 1 to phase 2. This is a clear evidence that an increase of the CO mixing ratio at the tropopause is not the cause for the observed lower stratospheric CO increase. This would be consistent with an increase of the fraction of young air of tropospheric origin and more efficient mixing as indicated in Fig. 6.

Panel (b) shows the distribution against equivalent latitude. Note that the observed increase is most prominent above $\Theta = 360$ K. This is a clear indication that mixing at $\Theta < 360$ K is suppressed due to the strong subtropical jet, which acts as a barrier for

mixing (Haynes and Shuckburgh, 2000) and would be consistent with enhanced mixing out of the TTL region.

## 5.2 Age spectra analysis

For further analysis of the relationship between diabatically descended, aged air with high transit times and potentially mixed young tropospheric air with low transit times we use age spectrum calculations of the CLaMS (Chemical Lagrangian Model

of the Stratosphere) (McKenna et al., 2002; Ploeger et al., 2015; Ploeger and Birner, 2016) model, which gives informations of the full transit time distribution. Notably we have the age spectral information for each individual data point along the flight



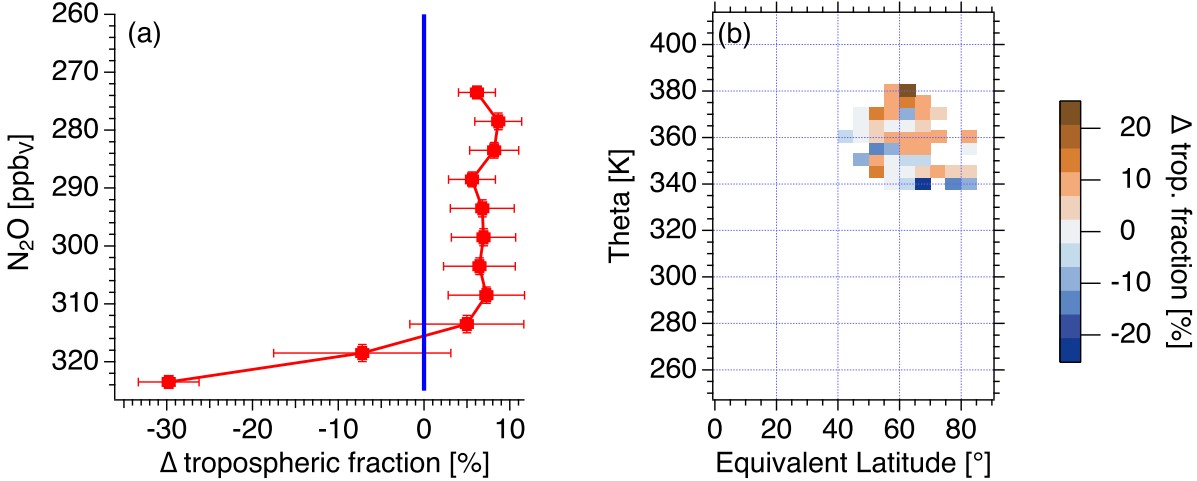

**Figure 8.** Panel (a): Tropospheric CO fraction from the mass balance equation as a function of $N_2O$ showing the difference between phase 2 and phase 1. Panel (b): The same as (a) but as distribution against equivalent latitude. Red colours indicate an increase of the tropospheric CO fraction.

track and therefore can directly compare our measurements with the spectrum.

To test if the model is able to reproduce the observations of tracers we compared CO and $N_2O$ from CLaMS with the measurements (Fig. 9). Model output is available along the flight track with a time resolution of ten seconds. Figure 9 shows the $N_2O$-CO tracer correlation for each data point. Panel (a) shows the correlation measured with the TRIHOP instrument, panel

(b) shows the correlation calculated out of the CLaMS model. As is evident CLaMS correctly represents the increase of CO relative to $N_2O$ from phase 1 to phase 2. Also the separate branches of the two phases are reproduced and the crossing of the correlation at 40 $ppb_V$ CO and 310 $ppb_V$ $N_2O$ is consistently simulated.

This remarkable agreement between model and observations further motivates the usage of CLaMS for age analysis of our measurements.

As mentioned before, CLaMS is able to calculate the full transit time distribution of analysed air masses along the flight track. Figure 10 shows the averaged age spectra of the CLaMS model for the respective phase (panel (a)) and the difference of them (panel (b)). Vertical dotted lines represent the mean age of the respective phase (blue and red) calculated by the CLaMS model, the dashed vertical lines separates young air masses with a mean age lower than 0.5 years and old air masses with mean age larger than 2 years. Since we have the full transit time distribution of each data point, we can compare this relation between

the different parts of the age spectrum. An increase of the tropospheric fraction would be linked to an increase of the part of the age spectrum with low transit times as indicated by the observed increase of CO relative to $N_2O$.

From Fig. 10 panel (b) it is evident that there is an increase of air masses older than two years up to 0.3%. In the range of air masses younger than six months, there is also an increase of the age spectrum between phase 2 and phase 1 evident which



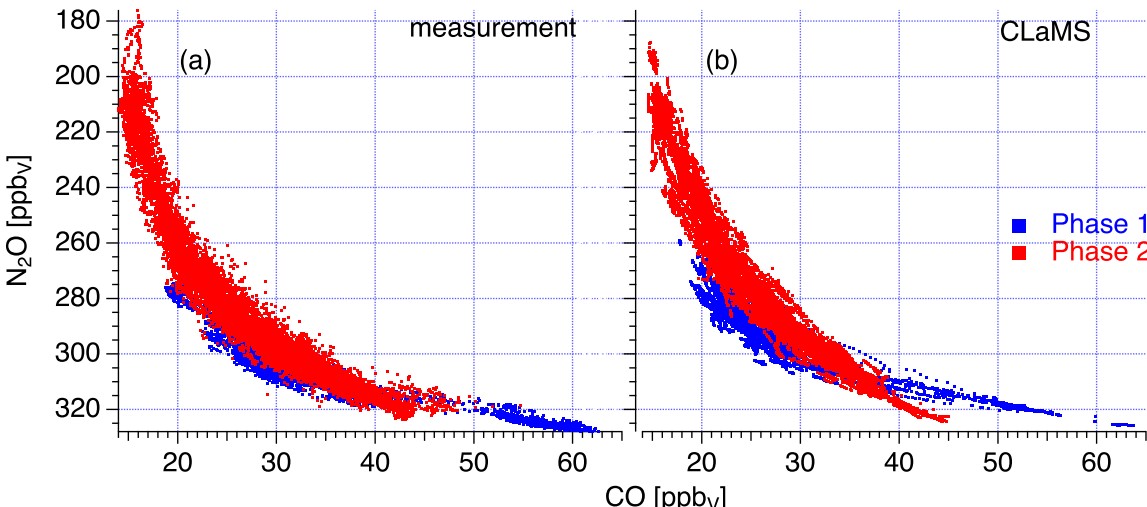

**Figure 9.** $N_2O$-CO correlation measured by the TRIHOP instrument (a) and CLaMS model output (b). Phase 1 coloured in blue, phase 2 coloured in red. The model output is available along the flight track with a time resolution of ten seconds.

is, with maximum values up to 0.9%, larger than the increase of the old air masses. The increase of the young fraction is in agreement with the observed increase of CO. It is therefore an indication for an increased mixing with air from the TTL at the end of winter.

Since the mean age is calculated as first moment of the distribution its value is most sensitive to changes by the old tail of the distribution (Hall and Waugh, 1997). Therefore the mean age rises by 0.27 years from 1.71 years to 1.98 years as a result of the increase of the age spectrum distribution for air masses older than two years. This matches the mean age increase of $SF_6$ and indicates, combined with the decrease of $N_2O$, the overall ageing in the lower stratosphere over the course of winter. Since the integral over the Greens function is normalised to one, the increases of air masses older than two years and younger than six months must result in a relative decrease in between. Therefore air masses with mean ages between 0.5 years and 2 years are more enhanced in phase 1 than in phase 2, which is evident by the change of the transit time distribution up to $-1.9\%$.

To further investigate the relationship of young versus aged air we used the spectral information for each individual data point in the following way. We calculated the accumulated fraction of air masses with transit times lower than six months and older than two years, respectively, for each data point. Figure 11 shows the binned fraction of air masses with transit times lower than six months versus the modelled mean age.

The comparison of the correlation for different times (phase 1 and phase 2) shows that for a given mean age a significant increase of the young tropospheric contribution is evident. Thus, according to the model and in agreement with the observed increase of CO, the late winter LS is more affected by tropospheric young air.

During winter 2015/16 CO mixing ratios in the LS increased from January to March while long-lived trace gases denote an





**Figure 10.** Panel (a): Averaged age spectra simulated by CLaMS for phase 1 (blue) and phase 2 (red). These spectra represent the mean of the individual age spectra available for each data point along the flight track. The mean age is indicated by the respective coloured vertical lines (phase 1: 1.71 years, phase 2: 1.98 years). The difference between both spectra is given in panel (b) showing an enhancement of both, young and old air masses from phase 1 to phase 2. Vertical dashed lines indicate the transit times of six months and 24 months, respectively (see next figures). The bin size of a data point is one month.





**Figure 11.** Mean age versus air fractions with transit times < 6 months from the age spectra simulated by CLaMS for phase 1 (blue) and phase 2 (red). Each data point is binned in steps of 5 ppb$_V$ N$_2$O. The variability in each bin is given by the vertical and horizontal lines, respectively.




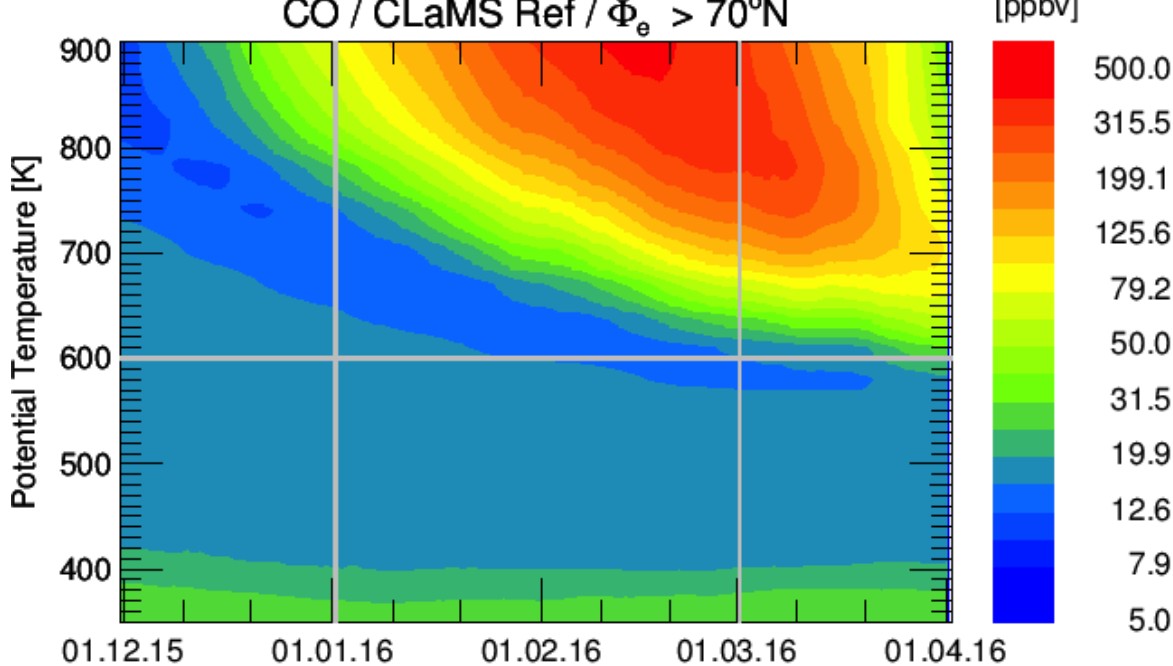

**Figure 12.** Temporal evolution of zonal mean CO for equivalent latitudes > 70 °N simulated by CLaMS for winter 2015/2016.

ageing of the LS. The analysis of CO-N$_2$O correlations, mass balance systems of transport pathways in the LS and model simulations points towards an increased influence of tropospheric air masses from the tropical lower stratosphere. Additional potential sources of CO in the LS are discussed in the following.

## 6   Discussion

5   Since there are different sources for CO at different locations in the atmosphere an increase of carbon monoxide mixing ratios can be due to (i) an increase of isentropic mixing out of the TTL, (ii) an increase of the tropospheric source strength, (iii) a potential influence of the mesosphere and (iv) a change of chemical reaction cycles due to higher amounts of reactive chlorine in the stratosphere. As already discussed the increase of enhanced tropospheric source emissions (ii) is highly unlikely (see Fig. 7). Since our analysis points to an increase of isentropic mixing out of the TTL (i), the possible influence of points (iii)
10   and (iv) have to be further discussed.

Carbon monoxide is produced in the mesosphere due to the photo-dissociation of carbon dioxide through high energetic sunlight. Therefore the composition of mesospheric air masses is clearly distinct from air mass composition of the stratosphere.





Rinsland et al. (e.g. 1999) found increased CO mixing ratios up to 90 ppb$_V$ at altitudes around 25 km or $\Theta = 630 - 670$ K and Engel et al. (2006b) found CO values of 600 ppb$_V$ at an altitude of 32 km. Both studies show very low N$_2$O mixing ratios (< 50 ppb$_V$). Although the authors found a layer of mesospheric air descending down to 22 km, this is not evident for the Arctic winter 2015/16 and lowest N$_2$O mixing ratios are found to be in the order of 200 ppb$_V$. CO mixing ratios simulated

by CLaMS in the chemistry setup are shown in Fig. 12 as vortex core average profile along the time throughout the winter and spring. Potential downward transport from mesospheric air is largest in the polar vortex core due to diabatic descent of air masses. With this, air with enriched CO mixing ratios is transported downward into the stratosphere over the course of winter. This is reflected in the MLS observations that determine the CLaMS upper boundary at $\Theta = 900$ K potential temperature. The simulation indicates the expected downward transport but sees no mesospheric influenced air masses in the LS.

It is evident that CO mixing ratios increase at $\Theta = 900$ K from December 2015 to end of February 2016 which descended from $\Theta = 900$ K down to $\Theta = 600$ K at the end of March 2016. Therefore these enhanced CO mixing ratios do not affect our measurement region below about $\Theta = 410$ K. For potential temperatures below $\Theta = 420$ K a slight increase in the CO mixing ratio with time is simulated, which does not originate in the stratosphere or mesosphere in agreement with our observations (see Fig. 7). Regarding to the composition of CO (Fig. 5) it is evident that the measured CO mixing ratios decrease with altitude

and the lowest values are found at the highest regions and equivalent latitudes. This indicates that the increase of measured CO mixing ratios has no mesospheric origin, because the enhanced CO mixing ratios are only transported down to $\Theta = 600$ K potential temperature in the CLaMS model.

Furthermore, an additional influence of descended mesospheric air into the lower stratosphere would not only impact CO but also N$_2$O. Due to the chemical differences between the stratospheric and the mesospheric composition, mixing of mesospheric

air, enriched in CO and depleted in N$_2$O, would lead to mixing lines very strongly differing from the observed relationship (see Fig. 6). Importantly, the CLaMS N$_2$O-CO correlation (Fig. 9) almost perfectly mirrors the observations, further indicating no mesospheric influence on the simulated correlation. Additionally the age spectrum calculations of the CLaMS model provide mass fractions of air masses regarding their stratospheric residence time. As is evident from Fig. 11 there is a significant increase of air masses younger than six months at typical mean ages for lower stratospheric air masses and mesospheric influence on

the basis of our analysis is highly unlikely.

In general, another important source of carbon monoxide in the atmosphere is the reaction of methane with reactive chlorine, which is not significant in the lower stratosphere (Flocke et al., 1999). Eventhough the influence of the methane reaction with Cl on CO is low, air masses enriched in reactive chlorine can be transported downwards, providing potentially reactants for the chemical production of CO. It may not be the case in this specific winter because of unprecedented low temperatures

and resulting higher chlorine activation. Therefore this aspect was also investigated in more detail. For this aim we simulated the CO yield from the reactions of CH$_4$ with chlorine, OH and O$^1$(D). To investigate the chemical sources and sinks of CO, CLaMS simulations in the boxmodel mode were performed. A large number of single air parcel backward trajectories starting on 15 March from locations within the vortex core (equivalent latitude > 65 °N; potential temperature between $\Theta = 350$ K and $\Theta = 500$ K) ending on 15 January and chemical composition changes were calculates using the CLaMS chemistry module

running forward in time for a subset of the trajectories with equivalent latitudes > 50 °N on 15 January (21480 trajectories).





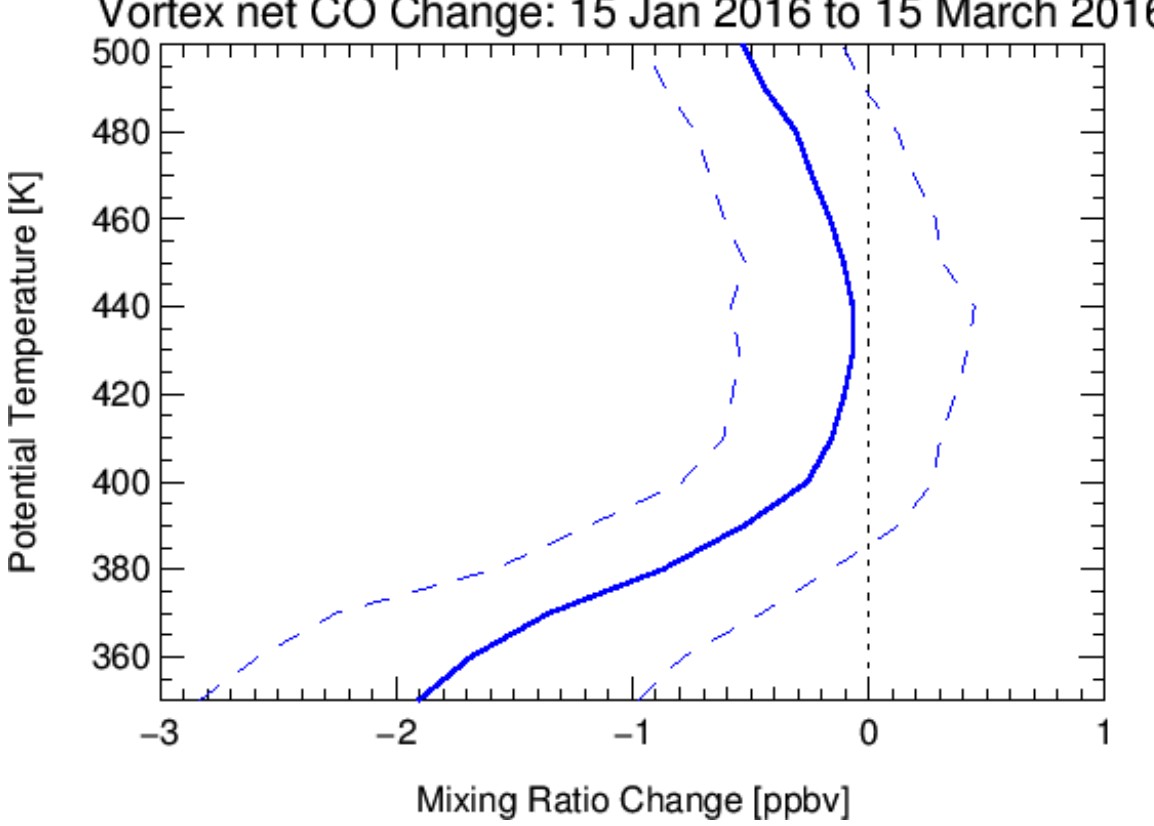

**Figure 13.** Net change of CO from January to March for air masses in the Arctic vortex (equivalent latitude > 65 °N) due to chemical reactions in the stratosphere and mesosphere calculated by CLaMS. The blue line represents the statistical mean, the dashed lines the 1-$\sigma$ standard deviation.

Figure 13 shows the statistical evaluation of the net CO change due to chemistry over the period as function of potential temperature on 15 March. The blue line represents the statistical mean and the dashed lines the 1-$\sigma$ standard deviation. As is evident the mean overall change is even negative over the entire profile, which is due to the oxidation of the produced CO by the reaction with OH. Therefore we conclude that the observed increase of CO in phase 2 is not due to the additional chemical

5  source reaction.

To investigate if transport and increased mixing of air mass fractions with transit times smaller than six months in winter 2015/16 was special compared to other years we analysed the climatology of these fractions from 2004 to 2016 and compared it to the calculated fractions in winter 2016, both from the CLaMS model (Fig. 14). The colour code represents the fractions of air masses with transit times smaller than six months, the contour lines represent the mean age and the thick black line indicates

10  the WMO tropopause. Note, that mixing of these air masses is significantly stronger depicted in the southern hemispheric polar





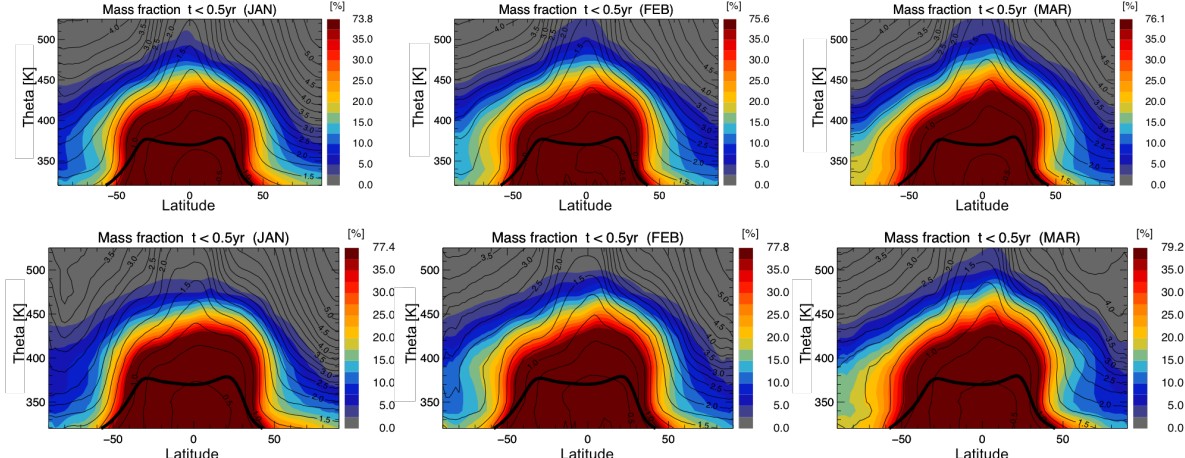

**Figure 14.** Zonal mean of air mass fractions (colour code) with transit times < 0.5 years for JFM from 2004 to 2016 climatology (upper row) and for 2016 only (lower row) against potential temperature. The contour lines show mean age in years and the thick black line the WMO tropopause.

region. This must not directly be compared to the northern hemisphere, since this time span represents summer months in the southern hemisphere, where mixing is larger compared to the winter.

From the climatology it is evident that the largest fraction of air masses with transit times smaller than six months exceeding 73% is found between 30 °S and 30 °N up to $\Theta = 430$ K. In January this strong signal has a sharp gradient at $\Theta = 450$ K. These air fractions are transported from January to March to the poles so that between 70 °N and 90 °N the fraction of air masses with transit times smaller than six months increases by 5 % at $\Theta = 380$ K. From January 2016 to March 2016 this transport is even stronger than in the climatology, as is evident from the convex structure of the distribution gradient to the north pole. From Fig. 14 it is also evident, that the mean age in March compared to January at $\Theta = 400$ K shows a simultaneous higher value in both, the climatology as well as the winter 2015/2016, whereas the structure of the mean age contours show a more horizontal meridional gradient in the winter 2016 compared to the climatology.

Finally, these findings show an enhancement of young fraction of air in the lower stratosphere of the polar Arctic region with an underlying increase of mean age of air indicating downward transported air masses of older air fractions. Both the enhanced transport of young air and the increased downwelling are evident from the climatology and turn out to be particularly strong in the winter 2016.

## 7  Summary

We present tracer measurements of CO and $N_2O$ measured during the POLSTRACC campaign in winter 2015/16 on board the German HALO research aircraft. The winter 2015/16 was characterised by an extreme cold and stable polar vortex which broke




up due to an MFW on 5. March 2016. In combination with measurements of $SF_6$ and model simulations by the CLaMS model it was possible to analyse the contributions of diabatic transport and isentropic mixing in the region of the upper troposphere / lower stratosphere. The mixing ratios of the long-lived trace gases $N_2O$ and $SF_6$ decrease over the course of winter and therefore denote an overall ageing due to subsiding air masses in the Arctic polar lower stratosphere. The calculated mean age shows for

$SF_6$ an ageing of 0.29 years and for CLaMS 0.27 years, respectively. Remarkably, the short-lived species CO increases at the same time. Since mixing can be identified by tracer-tracer correlations we used $CO$-$N_2O$ correlations to quantify the relation between transport and chemistry. Our analysis shows an increase by 3.7 $ppb_V$ CO relative to $N_2O$, which can be linked to an increase by 6.8 % of mixed air masses out of the TTL region. The comparison with the CLaMS model shows a very good agreement between measurements and model calculations. The $CO$-$N_2O$ correlation is well reproduced by the model. Analysis

of the averaged age spectrum for the respective phase shows that there is a simultaneous increase of fractions of air with transit times larger than two years and fractions of air with transit times smaller than six months. Since the mean age itself is most sensitive to changes on the old tail of the age spectrum, the ageing of air masses in the LS over the course of winter can be explained by the increase of old air masses, characterised by low $N_2O$ and $SF_6$ measurements. Increased mixing of young air masses adds to this and leads to an increased fraction of the younger part of the age spectrum, consistent with the observed

increase of CO. It is evident that this enhancement is due to stronger mixing processes out of the TTL region, where fresh tropospheric air is mixed into the polar lower stratosphere. Other potential sources of CO like mesospheric air and chemical reaction of $CH_4$ with chlorine are unlikely to have caused the observerd increase of CO.

Therefore we conclude that the Arctic lower stratosphere in March was strongly affected by mixing with young tropospheric air, which partly compensates for the overall ageing. These aged air masses are isentropically mixed with younger air masses

out of the TTL region. The observations are in-line with the climatology of mixing from 2005-2015 on the basis of Era-interim by the CLaMS model.

*Author contributions.* Jens Krause carried out the measurements and analysed the data with the help of Peter Hoor. Felix Plöger and Jens-Uwe Grooß did the model simulations with the CLaMS model. Andreas Engel, Harald Bönisch and Timo Keber provided the measurement data of $SF_6$ and mean age. Peter Hoor, Andreas Engel, Felix Plöger and Jens-Uwe Grooß provided helpful discussions and comments. Jens

Krause and Peter Hoor wrote the manuscript. Hermann Oelhaf, Björn-Martin Sinnhuber and Wolfgang Woiwode coordinated the POL-STRACC project.

*Acknowledgements.* This work was supported by the Deutsche Forschungsgemeinschaft (DFG, FKZ EN 367/13-1 and EN 367/11) and the Johannes Gutenberg-University Mainz (FKZ 8585084).

Jens Krause was partly funded under DFG grant HO 4225/7-1.

AGAGE is supported principally by NASA (USA) grants to MIT and SIO, and also by: DECC (UK) and NOAA (USA) grants to Bristol University; CSIRO and BoM (Australia): FOEN grants to Empa (Switzerland); NILU (Norway); SNU (Korea); CMA (China); NIES (Japan); and Urbino University (Italy).



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
