# Peer review of "Mixing and ageing in the polar lower stratosphere in winter 2015/2016"

_Atmospheric Chemistry and Physics, 2017_

## Referee Comment (RC1) · Anonymous Referee #1 · 2 Nov 2017

The paper uses aircraft tracer measurements from the POLSTRACC campaign combined with simulations with the CLaMS model to derive conclusions on the characteristics of transport in the high-latitude boreal lower stratosphere for the winter of 2015-16. The results show an increase of the mean age in the region from January to March 2016, which seems at odds with the increase in CO concentrations. The authors argue that this is due to a change in the age spectrum, which exhibits an increase in both old and young air by the end of the winter.

The article is well written and presents an interesting analysis motivated by in situ observations and nicely complemented with modeling tools. I recommend publication in ACP after the following few minor comments and technical corrections are addressed.

- The main point that should be addressed regards the high values of CO observed

during phase 1, seen in Fig. 5a at about 70 degrees and 330-340K, and in Fig. 7 (CO values above 45 ppbv). Although these values are not the main focus of the paper, they stand out, and there are a few parts in the paper where I miss some explanation of their origin. For instance, in the description on Page 13 Lines 25-29 it is mentioned that the 'direct tropospheric impact was greater in phase 1 than in phase 2' referring to these points. What do you mean by 'direct tropospheric impact'? Is this transport across the ExTL or did the high CO values originate in the TTL? Also in Section 5.2, you could look separately at the age of air spectrum for those air masses, instead of showing the results for all measurement points in phase 1 together. Does that help in interpreting the origin? Finally, some measurements in phase 1 were taken at lower latitudes (over Italy) compared to the rest of the campaign. Does that latitude difference have an impact on the CO values?

- P1 L22: 'diabatic descent [. . .] adds to the diabatic downwelling of the Brewer-Dobson circulation'. It seems to me you are referring to the same thing twice?

- P2 L29: tropical pipe

- P3 L23: The McPhaden reference is not about the 2015 ENSO event. A better option could be perhaps L'Heureux et al. (2017).

- P3 L24: The impact of the 2015-16 ENSO event on the polar vortez has been ana-lyzed by Palmeiro et al. (2017).

- P6. Only flights that were used for the analysis are shown.

- P8 L1: take → taken

- P8 L20: remove 'respective'

- P8 L27: remove respectively?

- P9 L7-8: is this a hypothesis or do you have an argument to support this statement?

- P9 L14: Are the physical altitude ranges the same for both phases?

- P9 L18: The mean increase of 0.29 is just below the precision of the mean age estimate from SF6. Do you still consider it a robust change?

- P9 L27: "with an variability" → an interannual variability?

- P9 L3: chapter → section

- P12 L4: [...] general picture of enhanced downwelling of the Brewer-Dobson circulation [...]

- P12 L9: despite the

- P13 L4-13: Could you refer to the individual panels of Fig. 6 as you describe the figure?

- P13 L21: an CO → a CO

- P13 L25-29: This description is unclear. What you mean by 'shows higher CO relative to N2O?'. Perhaps it would make it easier to follow if you referred to the isentropic levels approximately corresponding to the N2O values when you describe Fig. 7 (it is hard for the reader to combine mentally Figs. 4, 5 and 7).

- P17 L4-5: would it be more accurate to refer to these figures as 'scatter plots' rather than 'correlations'? Also on Fig. 9 caption.

- P17 L18: remove 'which is'

- P18 L8: Green's function

- P21 L1: what do you mean by 'mass balance systems of transport pathways'?

- P22 L27: Eventhough → Although

- P24 L8-10: This sentence is unclear. Do you mean that the high fraction of young air reaches higher latitudes in 2015/16 as compared to the climatology? If so, what is the variability (e.g. standard deviation) around the climatology? Is this winter statistically different from the climatology?

References:

L'Heureux, M.L., K. Takahashi, A.B. Watkins, A.G. Barnston, E.J. Becker, T.E. Di Liberto, F. Gamble, J. Gottschalck, M.S. Halpert, B. Huang, K. Mosquera-Vásquez, and A.T. Wittenberg, 2017: Observing and Predicting the 2015/16 El Niño. Bull. Amer. Meteor. Soc., 98, 1363–1382, https://doi.org/10.1175/BAMS-D-16-0009.1

Palmeiro, F. M., M. Iza, D. Barriopedro, N. Calvo, and R. García-Herrera (2017), The complex behavior of El Niño winter 2015–2016, Geophys. Res. Lett., 44, 2902–2910, doi:10.1002/2017GL072920.

---

## Referee Comment (RC2) · E. Ray (Referee) · 21 Nov 2017

This paper uses in situ aircraft measurements and the CLaMS model to investigate the transport characteristics responsible for observed trace gas correlations in the polar lower stratosphere. This is nice work and really highlights the power of using model age spectra to better understand the causes of measured trace gas distributions in the stratosphere. The model does a reasonable job of reproducing the general features of the observed trace gases but the age spectra is what really explains why the features exist. The data and techniques are well described and the conclusion are well supported. My main comments are around the discussion of the tracer-tracer curves and the grammar, which could have used more work before submission. I suggest publication with consideration of the comments below.

[Figure]

Main comments:

Figures 3-5 and 8b: I suggest changing the x and y axis ranges to eliminate white space and make the features more visible. You could change the theta minimum to 290 K for instance.

Pg. 13: I think this discussion of mixing and Figure 6 needs to be clearer. In Line 7 it is stated that "stratospheric CO will relax towards its stratospheric equilibrium value". But that's not really how it works. CO has a chemical lifetime in the stratosphere so it's destroyed at a certain rate. In the absence of mixing or transport it will be completely destroyed. You should cite Minschwaner et al. (2010) here for the CO chemical lifetime discussion.

In panel (d) I would recommend extending the blue curve up to the Chi_meso point since there is a background correlation curve that connects the stratospheric to the mesospheric values.

Lines 23-24: In the discussion of Figure 7 it's not clear that it's remarkable CO is higher relative to N2O in phase 2 compared to phase 1. The old air in the vortex that has come from high altitudes is expected to have relatively low N2O and CO but is it expected that the correlation will remain constant, or that CO will be lower relative to N2O? I just don't think it's well established what the correlation should be and if it is that should be justified by prior work.

Lines 24-25: This sentence is too vague to understand what it is referring to.

Lines 26-28: What does the "direct tropospheric impact" mean? This sentence should also be clarified.

Figure 14: I'd suggest making these plots NH only to see the features and differences in the region of interest more clearly. It would also be interesting to see line plots at 350K and 400K for example of mass fraction vs. latitude for climatology and 2016.

Grammar comments:

[Figure]

Pg. 2, line 6: "these air masses", what air masses are you referring to? Be more specific.

Pg. 2, line 23: comma needed after "vortex"

Pg. 2, line 24: comma needed after "result"

Pg. 2, line 25: "...establishes a relatively tropospheric..."

Pg. 3, line 9: "...conditions existed due to..."

Pg. 3, line 11: replace "was" with "were"

Pg. 3, line 13: replace "to" with "on"

Pg. 3, line 15: comma after second "warming"

Pg. 3, lines 19-21: be consistent with use of either "eastward" and "westward" or "easterly" and "westerly"

Pg. 3, line 24: "...El Nino could have accounted for a..."

Pg. 3, line 28: comma after "TTL"

Pg. 4, line 12: remove "the"

Pg. 4, line 13: remove "the aim of"

Pg. 4, line 16: remove "about"

Pg. 4, line 18: replace "of" with "that measured"

Pg. 4, line 22: add "and" between N2O and CO

Pg. 8, line 1: change "take" to "taken"

Pg. 8, line 2: change to "Green's"

Pg. 8, line 4: "...allows the calculation of time..."

[Figure]

Pg. 8, line 9: change "formation" to "formulation"

Pg. 8, line 10: "box model"

Pg. 8, line 21: remove "respective"

Pg. 8, line 27: change "constitute" to "contribute" and remove ", respectively"

Pg. 8, lines 30-31: "...mean age from long-lived tracer measurements, the tracer must have a..."

Pg. 9, line 26: change "the last" to "recent"

Pg. 9, line 27: remove "an"

Pg. 11, line 9: change "to" to "with"

Pg. 11, line 15: not all of the CO decreases below 360 K.

Pg. 11, line 16: change "rise" to "make"

Pg. 11, line 17: add "the" before "winter"

Pg. 12, line 5: "...with air from the tropical lower stratosphere."

Pg. 12, line 9: change "of" to "the" and "as" to "of", "...this increase originated..."

Pg. 12, line 10: "...TTL, into the extratropical lower stratosphere."

Pg. 12, line 11: add comma after "tropopause"

Pg. 12, line 13: "...as a stratospheric...", "used here as a tropospheric..."

Pg. 12, line 17: "effects"

Pg. 13, line 5: remove "actual"

Pg. 13, line 6: "...correlation is established..."

Pg. 13, line 26: remove "to" before "the"

Pg. 13, line 32: add a comma after "before"

Pg. 13, line 35: "...is the main source..."

Pg. 16, line 18: change "by" to "in"

Pg. 16, line 19: does the (3.7) refer to the uncertainty?

Pg. 16, line 30-31: "...information on the..."

Pg. 17, line 12: solid lines, not dotted lines Pg. 18, line 4: add "the" after "as", change "by" to "in"

Pg. 22, line 5: "...average profiles throughout..."

Pg. 22, line 6: change "from" to "of"

Pg. 22, line 14: remove "to", add comma after "(Fig. 5)"

Pg. 22, line 27: "even though"

Pg. 22, line 28: change "potentially" to "potential"

Pg. 22, line 30: add comma after "Therefore"

Pg. 22, line 32: "box model"

Pg. 22, line 34: "calculated"

Pg. 23, line 1: add "a" after "as"

Pg. 25, line 3: "decreased"

Pg. 25, line 4: "denoted"

Pg. 25, line 17: "observed"
* * *

---

## Referee Comment (RC3) · Anonymous Referee #3 · 5 Dec 2017

This paper analyzes aircraft observations of N2O, SF6, and CO from a recent polar aircraft campaign to show that the Arctic lowermost stratosphere has diabatic descent between January and March (this is well known, not new), and that while the mean age increases, CO measurements indicate that younger air mixes into the vortex during winter. A trajectory model simulation of the same period is used to support their interpretation of the measurements. The main result presented is that when a region of the atmosphere gets older, it doesn't mean that younger air didn't mix in! That is, there can be simultaneous changes in the age spectrum where both ends of the distribution change. The results show how 2 tracers with different lifetimes can be used to identify simultaneous contributions from young tropospheric and older stratospheric air. This isn't exactly how the message is stated but this is the interesting result.

[Figure]

The analysis of the aircraft data set is fine, the results are not surprising; they are consistent with what we know about the BDC and horizontal mixing in the LMS. Because the results are consistent with expectations, please present the data and analysis more succinctly.

The paper is far too long for what it has to say and it is not clear what the motivation is for this study. There is too much rudimentary discussion that is inappropriate for a journal article; some of the CO production and loss processes discussed are not actually significant or relevant. It feels like I am reading a thesis chapter not a journal article.

There are minor scientific issues regarding assumed CO values from the TTL and the assumption that CO in the lowermost stratospheric vortex is coming from the TTL. The assumed CO values are out of date (too low) and ignore copious recent data from MLS. There is large seasonal variability in tropical CO due to biomass burning (i.e., the CO tape recorder). Revisions should be made with up to date CO and this will change the results. I am not your thesis advisor but I would like to help you produce an article that others will be interested to read. There are many suggestions below intended to improve the readability of this article. This study is worth publishing after considerable shortening. This paper has 14 figures and well over 7000 words, but there is one main result.

General science and writing suggestions

The BDC is driven by waves. Wave energy deposited in the polar region warms the stratosphere which then radiatively relaxes, resulting in diabatic descent of polar air. So please do not say (p. 1, line 22) that diabatic descent in the vortex ADDS to the BDC. It is PART of the BDC. This occurs several times in the paper.

It is very rare that it makes sense for a paragraph to have only 1 or 2 sentences. Please review your choices of paragraph breaks. The small paragraphs here generally belong in a nearby paragraph.

The words 'respective' and 'respectively' are misused in nearly every occurrence. Even when used correctly here they are not necessary. Please delete all uses of these words.

Regarding the use of passive voice, many science papers unfortunately use expressions like 'In Figure 3, it is evident that....'. A paper will be more concise and interesting to read with 'Figure 3 shows....'. I suggest eliminating passive voice wherever possible. In general, replace 'it is evident that' with 'shows'. And 'it is possible to analyse' with 'we analyzed'.

Line by line

Introduction. What's missing here is any motivation for this study. Why are you investigating these data sets? Why would a reader want to continue reading?

p. 1, line 7: "We find an increasing influence..." Where?

p. 1, lines 22: diabatic descent does not add to the diabatic downwelling of the BDC. It is a part of the BDC. Same issue on p.2, line 7 and line 32. 'strong diabatic descent...AND the wave-driven BDC'. Again, these are one and the same thing. And later, mixing caused by breaking planetary waves is not 'in addition to' the BDC. Breaking planetary waves are what drives the BDC.

p. 2, line 3, 4: 'deep stratospheric air masses' Unclear. Can you be more specific than 'deep'? Define the LMS. Yes it's in the figure, but say here the latitudes, pressures, thetas, etc.

p. 2, line 6: The first sentence of the paragraph refers to 'these' air masses. Which ones?

p. 2, line 29: there is only a tropical pipe, no subtropical one.

p. 2, line 34. The reason H2O and O3 are modified by this transport is because they have large gradients. This is the underlying idea you might want to mention.

p. 3, lines 1-4. These 2 sentences about radiative effects come out of nowhere. What do they have to do with the rest of the introduction? Do you ever discuss radiative effects again in the paper? (I don't think so.) Either integrate these sentences into the context of this introduction or delete them.

p. 3, line 5. Define what you mean by subvortex. The vortex extends below 410K, so this is unclear.

p. 3, lines 8-13. The coldest early Arctic winter since 1948? What records do you use that go back to 1948 (no reference given). In looking at the met statistics since 1979, I see there is the occasional DAY in December 2015 that breaks a record, but that's all. This statement needs to be backed up with a reference and needs to be more specific in what way it was the coldest. Since the vortex extends below 400K, it doesn't make sense to say that the chemical influence of the vortex is seen below 400K.

p. 3, line 14 and beyond. I thought the commonly used expression was 'stratospheric final warming' (SFW), not MFW. Is there a reference for the statement about MFWs being rare earlier than mid-March?

p. 3, line 22. The QBO phase impact on vortex strength is only true for the northern hemisphere (Baldwin et al., 2001).

p. 3, lines 23-24. Suggested rewrite: "Matthias et al. [2016] argue that the strong El Nino weakened the 2016 Arctic vortex, but this is still under debate." (Same meaning using half the words. When sentences are short and to the point, the paper becomes more interesting to read.)

p. 4, line 14. Spell out what PVU stands for. And what are your PV units? Yours seem to be a factor of 10 smaller than in other papers. That's not a problem as long as you define your units.

p. 4, lines 24-29: 'Resolution of 10 seconds' and 0.1 Hz are the same thing. Infrared is one word.

p. 4, line 30 and beyond. Reword. "...operating between wavenumbers X and Y that measures CO, N2O, and CH4. The instrument uses a multi-pass White cell with a constant pressure of 30 hPa to minimize pressure broadening of the absorption lines. All measurements were integrated for 1.5 seconds." And later "In-flight calibration is performed against compressed ambient air standards that were calibrated against primary standards before and after the campaign. The primary standards are connected to..."

p. 7, lines 17-20. A better way to describe this would be in terms of the spatial resolution you have after the averaging. So if you are smoothing over a 20 minute period, then based on the aircraft speed, what spatial resolution do you end up with?

p. 8, lines 6-7. Suggested "A Clams simulation with full stratospheric chemistry was integrated as described by Grooss et al (2016)." Why does it matter that it is typically run for 6 months? Delete this unless you explain why this matters. p. 8, line 9. 'formulation' not formation.

p. 8, line 17. Why are the gravity wave flights excluded?

p. 8, lines 22-23. Equivalent latitude is not just 'linked' to PV, it is derived from PV. This paper concerns itself with diabatic descent in the vortex, but you note that PV is conserved under adiabatic conditions (true), and clearly the polar vortex is not adiabatic. How to you justify the use of PV? (The answer lies in the time scales for radiative relaxation.)

p. 8, line 27. Suggest "The transport pathways create an age spectrum or transit time distribution."

p. 9, lines 3-8. Which lifetime did you use for SF6 in your model? Another rewrite: "Models and observations both show a high bias of up to 1 year in the polar vortex." At what altitude in the vortex does this apply to?

p. 9, line 9-10. This is the only correct usage of 'respectively' I saw. Still, it's not

necessary because the 'respectively' is implied.

p. 9, lines 21-31. Way too much detail on N2O. Instead of 'distinct background value' it's really that it has a near constant value throughout the troposphere that makes strato-spheric influence identifiable. I would delete everything between "The tropospheric background value of N2O..." and "...Ko et al (2013)." You've already said the sources are at the surface, so delete the last 2 sentences of this paragraph and add: "N2O has a weak negative gradient above the tropopause that strengthens in winter due to increased diabatic descent."

Paragraph beginning p. 9, line 32. A clearer way to say this: "Figures 4a and 4b show N2O values between 276-325 ppbv were measured during Phase 1 and values below 200 ppbv were measured during Phase 2 above 400 K. Figure 4c shows an overall decrease in N2O in the polar LS due to diabatic descent during winter, consistent with mean age changes (Fig. 3c)."

Section 4.1.3 CO, first paragraph. This has way too much detail. Delete the 2nd sen-tence. 3rd sentence: "Due to the high variability of anthropogenic surface emissions, CO mixing ratios in the northern hemisphere vary from 70-200 ppbs [prinn] and the CO lifetime is on the order of weeks." 4th sentence ok. 5th & 6th sentences: "The main sink is oxidation by OH. The CO lifetime during polar night is a few months." Don't need reaction (1).

The phrase 'equilibrium value' is not right – this is not about chemical equilibria. I believe you are trying to describe a minimum stratosphere CO that comes from being far above tropospheric CO sources but well below where high mesospheric CO can influence the lower stratosphere. Please delete usage of equilibrium CO (happens a lot on p. 13). Also, 'stratospheric background value' used on the next page and elsewhere is a vague expression. Can you say what you mean by this?

p. 10, line 15. All you need to say is "The reaction of CH4 with Cl is an insignificant source of CO in the lower stratosphere [Flocke]."

p. 11, line 6. Figure 7 is mentioned before Figure 6. Reorder.

p. 11, line 10. The CO change during winter at 340 K is not so consistent with N2O. The CO changes small and are both positive and negative in this region. The results are mixed, not clear.

p. 11, line 15. Change 'rise the hypothesis' to 'suggest'

p. 11, line 17. Since there are no CO surface observations used, you can't really say anything about the strength of tropospheric source emissions.

p. 12, lines 2-5. I don't agree with the statement that the 'lowest values' are found in the 'furthest regions from the troposphere'. The difference plots (3c and 4c) show a range of differences (even a few increases), so if you were to quantify this (rather than showing color blocks) it's not obvious that you statement would be true. Can you be more quantitative, or at least modify the words here?

p. 12, line 4. 'contradicting' should be replaced, perhaps with 'unexpected'.

p. 12, lines 8-16. Here – and in general – there are too many sentences telling the reader what you are going to write about (or discuss later) rather than just writing about the subject. CO's use as a tracer has already been mentioned (Fig. 5) and the insignificant source form CH4 + CO has already been noted. Delete.

p. 13. Line 1-2. Use active voice as much as possible, e.g., 'tropospheric data have high N2O [how high?] and are accompanied by high CO while stratospheric data have N2O < 328 ppb'. p. 13, line 7. N2O is 'longer-lived' not 'chemical inert'.

p. 13, line 13. Regarding the line showing meso CO on Fig. 6d, it should point to a value of 0 N2O because there is essentially no N2O in the mesosphere.

p. 13, lines 23-28. This result is not so mysterious and it would not seem so if this were discussed in terms of the species' gradients. The changes observed during winter come from 2 processes: descent and mixing. The change in each tracer depends

on the tracer's vertical and horizontal gradients as well as the balance between vertical and horizontal motions. This is a more physically meaningful way to explain the observations.

p. 13, line 31. Try instead 'The region measured during both phases is 340-380K.'

p. 13 line 34-35. Why do you assume that the increased tropospheric fraction comes from the TTL? The POLSTRACC data didn't sample air south of 40N, so how can you be so certain of its origin? Isentropic mixing across the LMS tropopause (320-360K range) between the midlatitude UT and the polar LS would have the same effect, right? Without observations or a trajectory analysis, there isn't convincing evidence presented that the TTL is the source.

p. 16. Equations 2, 3, and 4 use mixing rations and fractions. I don't see any use of mass here. Why then is this described as mass balance?

p. 16, line 15. Why are you citing 10-20 year old studies for the value of CO at the tropical tropopause? You could be using something far more precise and current by looking at MLS CO measurements (2004-present, as low as $\sim$300 hPa). CO at the tropical tropopause has considerable seasonal variation due to seasonal biomass burning influence. MLS shows CO above 100 ppb in the TTL… much higher than what is used in your study – see Huang et al. [ACP, 2016]. This assumed TTL CO value impacts your results. This section needs to be revised based on recent measurements of TTL CO.

p. 16, lines 28-29. Should be 'longer transit times' and 'mixed with young tropospheric air with shorter transit times…'

p. 17, line 17. Try 'Figure 10b shows there is….' 0.3% is a very small change. What is the uncertainty on this number? Is 0.3% statistically different from zero?

p. 18, line 2. Try '…the observed CO increase, indicating increased mixing with air…'

p. 18, line 7. This is really the lowermost not the lower stratosphere you are talking

about.

p. 18, lines 11-12. Try 'To further investigate the relationship of young versus aged air, we calculate the accumulated fraction. . .'

p. 18. The paragraph beginning on line 15 should be part of the previous one.

p. 21, line 1-3. You could make a good point here that mean age is an incomplete or oversimplified way to characterize air. The age spectrum shows there changes in the contributions from older and younger age masses that determine the change in mean age.

p. 21, lines 5-10. You can cite Rinsland 1999 and Rosenfield 1994 that mesospheric CO just doesn't go this low into the stratosphere, and then you are done with this issue. The CH4+CL sources of CO has already been labeled insignificant. There is no need to talk about these insignificant sources again. This means that all of Section 6 (Discussion) before about p. 23 can be deleted.

p. 23, 10. Here's a suggestion for Fig. 14. Since you say that the SH results should not be compared to the NH, then don't show them. Figure 14 would be more readable if you just showed the regions of interest. Try making a 4-panel figure (Jan and March), 0-90N or 30-90N only.

p. 25, line 5. What are the significant figures here? Is 0.29 really different from 0.27, or are they both 0.3?

p. 25, line 18. The conclusions should be about the lowermost, not lower stratosphere as all the results are below 380K.

---

## Author Comment (AC2) · 23 Feb 2018

We thank Eric Ray for her/his comments and careful reading and changed the manuscript according to her/his suggestions. Our response is formatted as follows:

**Referee's comments**

**Author's reply**

**Changes to the manuscript**

This paper uses in situ aircraft measurements and the CLaMS model to investigate the transport characteristics responsible for observed trace gas correlations in the polar lower stratosphere. This is nice work and really highlights the power of using model age spectra to better understand the causes of measured trace gas distributions in the stratosphere. The model does a reasonable job of reproducing the general features of the observed trace gases but the age spectra is what really explains why the fea- tures exist. The data and techniques are well described and the conclusion are well supported. My main comments are around the discussion of the tracer-tracer curves and the grammar, which could have used more work before submission. I suggest publication with consideration of the comments below.

**Main comments:**

 Figures 3-5 and 8b: I suggest changing the x and y axis ranges to eliminate white space and make the features more visible. You could change the Θ minimum to 290 K for instance.

**Changed**

- Pg. 13: I think this discussion of mixing and Figure 6 needs to be clearer. In Line 7 it is stated that "stratospheric CO will relax towards its stratospheric equilibrium value". But that's not really how it works. CO has a chemical lifetime in the stratosphere so it's destroyed at a certain rate. In the absence of mixing or transport it will be completely destroyed. You should cite Minschwaner et al. (2010) here for the CO chemical lifetime discussion.

With the term 'equilibrium value' we refer to the equilibrium between CO production from methane oxidation (CH4+OH is the rate limiting step) and the degradation of CO via OH. In the lower stratosphere there are two important reactions which determine the abundance of CO. The first one is the production of CO via methane oxidation and OH, the second one is the sink reaction of CO with OH to  $CO_2$ , which is much faster than the production from methane. Since both reactions are driven by OH, the CO concentration in the lower stratosphere depends on the available OH and methane. In the absence of transport from the troposphere and subsequent mixing CO will be degraded rapidly, but not to zero since the CO production from methane, which has a long lifetime in the stratosphere of (190 ± 50) years (Brown et al., 2013) acts as a source as long as methane and OH are available. A steady state equilibrium is the result according to:  $[CO]_eq = k1/k2 \ [CH_4] \ after solving \ d[CO]/dt = k1 \ [CH_4] \ [OH] - k2 \ [CO] \ [OH] = 0.$  Note that the value is independent from OH, since  $[OH] \ cancels \ out.$  Therefore in the long-term limit this leads to a CO value of 5-15 ppbv depending on the integrated temperature history of the air masses. We used the term 'equilibrium value' to emphasize, that we

refer to a chemically driven equilibrium. As long as methane is present in the stratosphere, CO can be produced.

Looking at tracer tracer correlations, the CO-equilibrium can be seen in vertical branches of correlations, when using CO as x-axis (see e.g. Fig. 9). This vertical branch indicates vanishing CO variability and has been observed e.g by Flocke et al. 1999 and Herman et al. 1999.

Therefore we kept the term 'CO-equilibrium value'.

The analysis of Minschwaner et al. (2010) focuses on the CO chemistry in the upper stratosphere and mesosphere with MLS measurements, where CO is enhanced from  $CO_2$  photolysis and no equilibrium exists. As can be seen in Fig.7 and Fig.12 of the original manuscript a region constant CO for  $N_2O < 220$  ppbv is evident, which indicates this constant backround CO\_eq.

- In panel (d) I would recommend extending the blue curve up to the Chi\_meso point since there is a background correlation curve that connects the stratospheric to the mesospheric values.

Changed. The figure now accounts for high CO values and zero  $N_2O$  in the mesosphere.

- Lines 23-24: In the discussion of Figure 7 it's not clear that it's remarkable CO is higher relative to N2O in phase 2 compared to phase 1. The old air in the vortex that has come from high altitudes is expected to have relatively low N2O and CO but is it expected that the correlation will remain constant, or that CO will be lower relative to N2O? I just don't think it's well established what the correlation should be and if it is that should be justified by prior work.

Given that the fraction of descending aged air depleted in  $N_2O$  and  $SF_6$  increases, one would expect at least not an increase in CO due to its much shorter chemical lifetime. Once CO is in steady state one would expect no sloped correlation at all (i.e. a vertical branch in Fig.7).

**Lines 24-25: This sentence is too vague to understand what it is referring to.**

**Manuscript changed to:**

It is important to note that the correlation along the mixing line which connects tropospheric values with the stratosphere shows higher CO relative to N2O in phase 2. As indicated in Fig. 6 this is a clear indication for enhanced mixing of tropospheric air masses at for N2O values > 273 ppbv.

**- Lines 26-28: What does the "direct tropospheric impact" mean? This sentence should also be clarified.**

The expression 'direct tropospheric impact' should indicate, that the increase of CO relative to  $N_2O$  occurs relatively close to the tropause at high (but stratospheric) values of  $N_2O$  (i.e. at most tropospheric influenced air masses). Here, air parcels which have been recently transported into the lowermost stratosphere have in general the shortest stratospheric residence time. It is not possible to derive from Figure 7 the information whether transport out of the TTL region occurred in this specific case or from the ExTL.

Therefore we changed the sentence to:

Therefore we can conclude that regarding the CO-N2O correlation the tropospheric impact on short timescales through the ExTL was greater in phase 1 than in phase 2, [...]

 Figure 14: I'd suggest making these plots NH only to see the features and differences in the region of interest more clearly. It would also be interesting to see line plots at 350K and 400K for example of mass fraction vs. latitude for climatology and 2016.

We changed figure 14 to a 4x4 plot of January and March for the northern hemisphere only.

The below graph R1 shows line plots of the relative difference of air masses with transit times smaller than six months (MF06) from March 2016 to March of the climatology (thick line) at 350 K and 400 K.

At latitudes northwards 60° there are up to 10% more MF06 air masses as compared to the climatology, which also supports our hypothesis.

Fig. R1: Line plot of difference (thick line) between March 2016 and March of the climatology. Dashed lines denote the standard deviation.

- Grammar comments:
- Pg. 2, line 6: "these air masses", what air masses are you referring to? Be more specific.

**Manuscript changed to:**

Air masses descending from the upper stratosphere and mesosphere chemically differ from the composition of the LMS, since they are potentially affected by ozone depleting catalytic cycles [...]

- Pg. 2, line 23: comma needed after "vortex"

Changed

- Pg. 2, line 24: comma needed after "result"

Changed

- Pg. 2, line 25: "... establishes a relatively tropospheric..."

Changed to the suggestion

- Pg. 3, line 9: ". . .conditions existed due to. . ."

Changed to the suggestion

- Pg. 3, line 11: replace "was" with "were"

Changed

- Pg. 3, line 13: replace "to" with "on"

**Changed**

- Pg. 3, line 15: comma after second "warming"

**Changed**

- Pg. 3, lines 19-21: be consistent with use of either "eastward" and "westward" or "easterly" and "westerly"

Changed to westerly and easterly

- Pg. 3, line 24: "... El Nino could have accounted for a..."

Changed to the suggestion

- Pg. 3, line 28: comma after "TTL"

Changed

- Pg. 4, line 12: remove "the"

Changed

- Pg. 4, line 13: remove "the aim of"

**Changed**

- Pg. 4, line 16: remove "about"

**Changed**

- **Pg. 4, line 18: replace "of" with "that measured"** *Changed to the suggestion*
- Pg. 4, line 22: add "and" between N2O and CO

**Changed**

- Pg. 8, line 1: change "take" to "taken"

**- Changed**

- Pg. 8, line 2: change to "Green's"

**Changed**

- Pg. 8, line 4: ". . .allows the calculation of time. . ."

**Changed**

- Pg. 8, line 9: change "formation" to "formulation"

**Changed**

- Pg. 8, line 10: "box model"

**Changed**

- Pg. 8, line 21: remove "respective"

**Changed**

- Pg. 8, line 27: change "constitute" to "contribute" and remove ", respectively"

**Changed**

- Pg. 8, lines 30-31: ". . .mean age from long-lived tracer measurements, the tracer must have a. . ."

Changed to the suggestion

- Pg. 9, line 26: change "the last" to "recent"

Changed to the suggestion

- Pg. 9, line 27: remove "an"

Changed

- Pg. 11, line 9: change "to" to "with"

Changed

- Pg. 11, line 15: not all of the CO decreases below 360 K.

Manuscript changed to:

Note that the main increase is observed above  $\Theta = 360$  K and  $50^{\circ}$  equivalent latitude. Below  $\Theta = 360$  K more areas with decreasing values are encountered until there is no left increase at  $\Theta = 340$  K.

- Pg. 11, line 16: change "rise" to "make"

Changed to the suggestion

- Pg. 11, line 17: add "the" before "winter"

Changed to the suggestion

- Pg. 12, line 5: ". . .with air from the tropical lower stratosphere."

Changed to the suggestion

- Pg. 12, line 9: change "of" to "the" and "as" to "of", ". . .this increase originated. . ."

Changed to the suggestion, but left the expression as the mesosphere, because the mesosphere is an example of one of the potential sources for stratospheric CO.

- Pg. 12, line 10: "...TTL, into the extratropical lower stratosphere."

Changed to the suggestion

- Pg. 12, line 11: add comma after "tropopause"

Changed

- Pg. 12, line 13: "...as a stratospheric...", "used here as a tropospheric..."

Changed

- Pg. 12, line 17: "effects"

Changed

- Pg. 13, line 5: remove "actual"

Changed

- Pg. 13, line 6: ". . .correlation is established. . ."

Changed to the suggestion

- Pg. 13, line 26: remove "to" before "the"

Changed

- Pg. 13, line 32: add a comma after "before"

Changed

- Pg. 13, line 35: "... is the main source..."

Changed

- Pg. 16, line 18: change "by" to "in"

Changed

- Pg. 16, line 19: does the (3.7) refer to the uncertainty?

Yes, changed to

**6.8 +- 3.7 %**

- Pg. 16, line 30-31: ". . . information on the. . ."

Changed

- Pg. 17, line 12: solid lines, not dotted lines

Changed

- Pg. 18, line 4: add "the" after "as", change "by" to "in"

Changed

- Pg. 22, line 5: ". . .average profiles throughout. . ."

Changed to the suggestion

- Pg. 22, line 6: change "from" to "of"

Changed

- Pg. 22, line 14: remove "to", add comma after "(Fig. 5)"

Changed

- Pg. 22, line 27: "even though"

Changed

- Pg. 22, line 28: change "potentially" to "potential"

Changed

- Pg. 22, line 30: add comma after "Therefore"

Changed

- Pg. 22, line 32: "box model"

Changed

- Pg. 22, line 34: "calculated"

Changed

- Pg. 23, line 1: add "a" after "as"

Changed

- Pg. 25, line 3: "decreased"

Changed

- Pg. 25, line 4: "denoted"

Changed

- Pg. 25, line 17: "observed"

Changed

**References:**

Brown, A. T., C. M. Volk, M. R. Schoeberl, C. D. Boone, und P. F. Bernath (2013). Stratospheric lifetimes of CFC-12, CCl4, CH4, CH3Cl and N2O from measurements made by the Atmospheric Chemistry Experiment-Fourier Transform Spectrometer (ACE-FTS). Atmos. Chem. Phys. 13(14), 6921–6950, doi:10.5194/acp-13-6921-2013.

K. Minschwaner, G. L. Manney, N. J. Livesey, H. C. Pumphrey, H. M. Pickett, L. Froidevaux, A.

Lambert, M. J. Schwartz, P. F. Bernath, K. A. Walker, and C. D. Boone, "The photochemistry of carbon monoxide in the stratosphere and mesosphere evaluated from observations by the Microwave Limb Sounder on the Aura satellite," J. Geophys. Res. Atmos., vol. 115, no. 13, pp. 1–9, 2010.

---

## Author Comment (AC3) · 23 Feb 2018

We thank Referee #3 for her/his comments and careful reading and changed the manuscript according to her/his suggestions. Our response is formatted as given below:

**Referee's comments**

*Author's reply*

Changes to the manuscript

**This paper analyzes aircraft observations of N2O, SF6, and CO from a recent polar aircraft campaign to show that the Arctic lowermost stratosphere has diabatic descent between January and March (this is well known, not new), and that while the meanage increases, CO measurements indicate that younger air mixes into the vortex during winter. A trajectory model simulation of the same period is used to support their interpretation of the measurements. The main result presented is that when a region of the atmosphere gets older, it doesn't mean that younger air didn't mix in! That is, there can be simultaneous changes in the age spectrum where both ends of the distribution change. The results show how 2 tracers with different lifetimes can be used to identify simultaneous contributions from young tropospheric and older stratospheric air. This isn't exactly how the message is stated but this is the interesting result.**

**The analysis of the aircraft data set is fine, the results are not surprising; they are consistent with what we know about the BDC and horizontal mixing in the LMS. Because the results are consistent with expectations, please present the data and analysis more succinctly.**

**The paper is far too long for what it has to say and it is not clear what the motivation is for this study. There is too much rudimentary discussion that is inappropriate for a journal article; some of the CO production and loss processes discussed are not actually significant or relevant. It feels like I am reading a thesis chapter not a journal article.**

**There are minor scientific issues regarding assumed CO values from the TTL and the assumption that CO in the lowermost stratospheric vortex is coming from the TTL. The assumed CO values are out of date (too low) and ignore copious recent data from MLS. There is large seasonal variability in tropical CO due to biomass burning (i.e., the CO tape recorder). Revisions should be made with up to date CO and this will change the results. I am not your thesis advisor but I would like to help you produce an article that others will be interested to read. There are many suggestions below intended to improve the readability of this article. This study is worth publishing after considerable shortening. This paper has 14 figures and well over 7000 words, but there is one main result.**

**General science and writing suggestions**

**The BDC is driven by waves. Wave energy deposited in the polar region warms the stratosphere which then radiatively relaxes, resulting in diabatic descent of polar air. So please do not say (p. 1, line 22) that diabatic descent in the vortex ADDS to the BDC. It is PART of the BDC. This occurs several times in the paper.**

*Changed accordingly throughout the manuscript.*

**It is very rare that it makes sense for a paragraph to have only 1 or 2 sentences. Please review your choices of paragraph breaks. The small paragraphs here generally belong in a nearby paragraph.**

*Changed wherever possible*

**The words 'respective' and 'respectively' are misused in nearly every occurrence. Even when used correctly here they are not necessary. Please delete all uses of these words.**
**Regarding the use of passive voice, many science papers unfortunately use expressions like 'In Figure 3, it is evident that ...'. A paper will be more concise and interesting to read with 'Figure 3 shows ....'. I suggest eliminating passive voice wherever possible. In general, replace 'it is evident that' with 'shows'. And 'it is possible to analyse' with 'we analyzed'.**

*We eliminated the frequent use of passive voice and inserted the suggested expressions. Note further, that remaining grammar and language issues will be handled by production office during the copy editing process.*

**Line by line**

**Introduction. What's missing here is any motivation for this study. Why are you investigating these data sets? Why would a reader want to continue reading?**

*We highlighted the motivation in the first sentence and changed the order of the introduction:*

Uncertainties in the description of mixing introduce large uncertainties to quantitative estimates of radiative forcing (Riese et al., 2012) which are on the order of 0.5 W m^-2 (Riese et al., 2012). Therefore it is important to quantify the contribution of the dynamical processes which act on the distribution of tracers. The arctic UTLS during winter is affected by diabatic descent from the stratosphere and quasi horizontal mixing by the shallow branch of the Brewer-Dobson circulation, which connects the tropical tropopause region with the high arctic (e.g. Rosenlof et al., 1997; Birner and Bönisch, 2011).
We present data from winter 2015/2016, which were measured during the POLSTRACC (The Polar Stratosphere in a Changing Climate) aircraft campaign between December 2015 and March 2016 in the Arctic upper troposphere and lower stratosphere (UTLS).
The focus of this work is on the role of transport and mixing between aged and potentially chemically processed air masses from the stratosphere with mid and low latitude air mass fractions with small transit times originating at the tropical lower stratosphere. By combining measurements of CO, $N_2O$ and $SF_6$ we investigate the evolution of the relative contributions of transport and mixing to the UTLS composition over the course of the winter.

**p. 1, line 7: "We find an increasing influence ..." Where?**

*We added a more precise description of the measurement region to the beginning of the abstract.*

We find an increasing influence of aged stratospheric air partly from the vortex as indicated by decreasing $N_2O$ and $SF_6$ values over the course of winter in the extratropical lower and lowermost stratosphere between $\Theta = 360$-$\Theta = 410$ K over the North Atlantic and the European Arctic.'

**p. 1, lines 22: diabatic descent does not add to the diabatic downwelling of the BDC. It is a part of the BDC. Same issue on**
**p.2, line 7 and line 32. 'strong diabatic descent the wave-driven BDC'. Again, these are one and the same thing. And later, mixing caused by breaking planetary waves is not 'in addition to' the BDC. Breaking planetary waves are what drives the BDC.**

*We changed the manuscript to the suggestion. The stratospheric circulation including the descent in the polar vortices is of course driven by the wave-driven Brewer-Dobson-circulation. The usage of the word ADD should not imply that there are two distinct processes resulting in downward transport of the polar winter stratosphere.*

During winter the UTLS-region (Fig. 1) at high latitudes is strongly affected by the evolution of the polar vortex. Diabatic descent in the polar stratosphere, which is strongest inside the polar vortex results as part of the Brewer-Dobson circulation (Brewer1949,Dobson1956) in mid and high latitudes as response to the breaking of planetary and gravity waves (Haynes1991,Plumb2002,Butchart2014) in the upper stratosphere and mesosphere. […] These air masses are rapidly mixed quasi horizontally by breaking planetary waves with descending aged air from high latitudes as part of the shallow branch of the Brewer-Dobson circulation (Birner2011,Abalos2013). […]

**p. 2, line 3, 4: 'deep stratospheric air masses' Unclear. Can you be more specific than 'deep'? Define the LMS. Yes it's in the figure, but say here the latitudes, pressures, Θs, etc.**

*Changed. We added definitions from literature, where available.*

This downwelling leads to an increasing contribution of stratospheric air masses from the overworld (defined as the region, where isentropes are entirely located in the stratosphere, Hoskins, 1991). Over the course of winter they contribute to the composition of the lower overworld ($\Theta < 420$ K), where our measurements took place, and the lowermost stratosphere (LMS) (Rosenfield et al.,1994) (defined as the region bounded by the 380 K isentrope and the extratropical tropopause (Rosenfield1994, Holton1995))

**p. 2, line 6: The first sentence of the paragraph refers to 'these' air masses. Which ones?**

*Manuscript changed to:*

Air masses descending from the upper stratosphere and mesosphere chemically differ from the composition of the LMS, since they are potentially affected by ozone depleting catalytic cycles (Solomon1999).

**p. 2, line 29: there is only a tropical pipe, no subtropical one.**

*Changed*

**p. 2, line 34. The reason H2O and O3 are modified by this transport is because they have large gradients. This is the underlying idea you might want to mention.**

*We want to refer to air masses, which have entered the stratosphere across the TTL region and the 380 K isentrope. We changed the sentence:*

Above $\Theta = 380$K these air masses, which ascended through the TTL […]

*and*

This rapid transport modifies the abundance of particularly water vapour and ozone in this region, which have seasonally varying isentropic gradients (e.g. Plöger et al.2013) […]

**p. 3, lines 1-4. These 2 sentences about radiative effects come out of nowhere. What**

**do they have to do with the rest of the introduction? Do you ever discuss radiative effects again in the paper? (I don't think so.) Either integrate these sentences into the context of this introduction or delete them.**

*We followed the suggestion and added a paragraph at the beginning of the introduction (see comment above related to the general motivation at the beginning of the introduction).*

**p. 3, line 5. Define what you mean by subvortex. The vortex extends below 410K, so this is unclear.**

*At the maximum flight ceiling of HALO, which was 15km during POLSTRACC, we couldn't reach the interior vortex, but frequently encountered large filaments of vortex air as evident from the trace gas composition. We use the term "subvortex region" for the isentropes below 420 K (approximately 120 hPa) where the lower vortex exists, but starts to break up at lower altitudes. Therefore we changed the sentence to:*

In our study we focus on the transition of the tracer composition in the vortex affected UTLS region up to […]

**p. 3, lines 8-13. The coldest early Arctic winter since 1948? What records do you use that go back to 1948 (no reference given). In looking at the met statistics since 1979, I see there is the occasional DAY in December 2015 that breaks a record, but that's all. This statement needs to be backed up with a reference and needs to be more specific in what way it was the coldest. Since the vortex extends below 400K, it doesn't make sense to say that the chemical influence of the vortex is seen below 400K.**

*As shown in Matthias et al., 2016, the winter 2015/2016 was extraordinary cold in November / December based on ERA Interim (1979-2015) and NCEP/NCAR reanalysis from 1948-2016. They state in the first sentence of their abstract: "The Arctic polar vortex in the early winter 2015/2016 was the strongest and coldest of the last 68 years."*

*We therefore changed the sentence accordingly:*

The early Arctic winter 2015/16 (November/December) was among the coldest winters in the lower stratosphere (LS) since 1948.

**p. 3, line 14 and beyond. I thought the commonly used expression was 'stratospheric final warming' (SFW), not MFW. Is there a reference for the statement about MFWs being rare earlier than mid-March?**

*We have used the term 'major final warming' (MFW) from Manney and Lawrence, 2016, and added the reference.*

**p. 3, line 22. The QBO phase impact on vortex strength is only true for the northern hemisphere (Baldwin et al., 2001).**

*Baldwin et al (2001) state that they found an impact for the NH polar vortex and that the NH is more sensitive to breaking of planetary waves, due to larger wave amplitudes and disrupted circulation pattern by major warmings. Therefore we changed the manuscript:*

Since the QBO affects the zonal wind direction in the tropical lower stratosphere (Niwano et al., 2003) its strength and phase is crucial for stratospheric transport processes (Baldwin et al., 2001) and westerly phases are related to strong and cold polar Arctic vortex.

**p. 3, lines 23-24.** Suggested rewrite: "Matthias et al. [2016] argue that the strong El Nino weakened the 2016 Arctic vortex, but this is still under debate." (Same meaning using half the words. When sentences are short and to the point, the paper becomes more interesting to read.)

*Changed to the suggestion.*

**p. 4, line 14.** Spell out what PVU stands for. And what are your PV units? Yours seem to be a factor of 10 smaller than in other papers. That's not a problem as long as you define your units.

*Changed.*

$1 \text{ PVU} = 10^{-6} \text{ m}^2 \text{ s}^{-1} \text{ K kg}^{-1}$

**p. 4, lines 24-29:** 'Resolution of 10 seconds' and 0.1 Hz are the same thing. Infrared is one word.

*Changed.*

**p. 4, line 30 and beyond.** Reword. "... operating between wavenumbers X and Y that measures CO, N2O, and CH4. The instrument uses a multi-pass White cell with a constant pressure of 30 hPa to minimize pressure broadening of the absorption lines. All measurements were integrated for 1.5 seconds." And later "In-flight calibration is performed against compressed ambient air standards that were calibrated against primary standards before and after the campaign. The primary standards are connected to..."

*The manuscript is changed to:*

*The TRIHOP instrument (Schiller2008) is an infrared absorption laser spectrometer with three quantum cascade lasers (QCL) operating between wavenumbers $1269 \text{ cm}^{-1}$ and $2184 \text{ cm}^{-1}$ that measures CO, $N_2O$, and $CH_4$. The instrument uses a multi-pass White cell with a constant pressure of 30 hPa to minimize pressure broadening of the absorption lines. All measurements were integrated for 1.5 seconds. The three species are subsequently measured during a full cycle which finally leads to a time resolution of 7 seconds due to additional latency times when the channels are switched. In-flight calibration is performed against compressed ambient air standards that were calibrated against primary standards before and after the campaign. The primary standards are tracable to the World Meteorological Organisation Global Atmosphere Watch Central Calibration Laboratory (WMO GAW CCL) scale (X2007) for greenhouse gases. During POLSTRACC it was possible to achieve a (2 sigma) precision of CO, $N_2O$ and $CH_4$ of 1.15, 1.84 and 9.46 $ppb_V$ respectively.*

**p. 7, lines 17-20.** A better way to describe this would be in terms of the spatial resolution you have after the averaging. So if you are smoothing over a 20 minute period, then based on the aircraft speed, what spatial resolution do you end up with?

*Note that the $SF_6$ data are not smoothed over 20 minutes. A correlation fit to the CFC-12 is used for each individual $SF_6$ data point with an interval length of 20 minutes. Since the slope of the $SF_6$-CFC-12 correlation is robust and both species are affected by atmospheric variability in*

*the same way, this procedure reduces the noise without smoothing atmospheric variability. Since the data output of the GhOST-MS instrument is one minute, the spatial resolution of the SF$_6$ measurements is 15 km; see page 4, line 26.*

**p. 8, lines 6-7. Suggested "A Clams simulation with full stratospheric chemistry was integrated as described by Grooss et al (2016)." Why does it matter that it is typically run for 6 months? Delete this unless you explain why this matters.**

*Sentence changed to the suggestion.*

A CLaMS simulation with full stratospheric chemistry was integrated as described by Grooss et al. (2014). The upper boundary is set to Θ = 900 K potential temperature, […]

**p. 8, line 9. 'formulation' not formation.**

*Changed*

**p. 8, line 17. Why are the gravity wave flights excluded?**

*Since gravity waves introduce variability and turbulence which are not covered by the PV fields of the meteorological analysis the consistency of equivalent latitude and tracer observations would be destroyed.*

**p. 8, lines 22-23. Equivalent latitude is not just 'linked' to PV, it is derived from PV. This paper concerns itself with diabatic descent in the vortex, but you note that PV is conserved under adiabatic conditions (true), and clearly the polar vortex is not adiabatic. How to you justify the use of PV? (The answer lies in the time scales for radiative relaxation.)**

*Radiative time scales are long (on the order of weeks) since cooling rates in the vortex are typically 0.5 K/day. In contrast adiabatic excursions of the tropopause by Rossby Waves occur on timescales of days, which allows to separate tropospheric and stratospheric air masses by using equivalent latitude for large scale motions. Small scale non-conservative (i.e. non isentropic) processes lead to deviations between PV and comparing tracer and PV.*

**p. 8, line 27. Suggest "The transport pathways create an age spectrum or transit time distribution."**

*We changed the sentence to your suggestion.*

**p. 9, lines 3-8. Which lifetime did you use for SF6 in your model?**

*CLaMS does not use SF$_6$ to calculate mean age. Therefore we added a sentence to the CLaMS section:*

Mean age in CLaMS is calculated from an inert model "clock-tracer" with linear increasing mixing ratio at the surface (Hall et al., 1994). The resulting mean ages are fully consistent with mean age calculated as the first moment of the CLaMS age spectrum (Ploeger and Birner, 2016).

**Another rewrite:**
**"Models and observations both show a high bias of up to 1 year in the polar vortex."**

*Changed*

**At what altitude in the vortex does this apply to?**

*It is impossible to give a unique number here, since this critically depends on the descent and mixing in the vortices and may therefore vary from winter to winter. For 2015/2016 the effect might have occurred down to Θ = 550K or 600 K (see Fig. 12)*

**p. 9, line 9-10. This is the only correct usage of 'respectively' I saw. Still, it's not necessary because the 'respectively' is implied.**

*'Respectively' removed*

**p. 9, lines 21-31. Way too much detail on N2O. Instead of 'distinct background value' it's really that it has a near constant value throughout the troposphere that makes stratospheric influence identifiable. I would delete everything between "The tropospheric background value of N2O ... " and " ... Ko et al (2013)." You've already said the sources are at the surface, so delete the last 2 sentences of this paragraph and add: "N2O has a weak negative gradient above the tropopause that strengthens in winter due to increased diabatic descent."**

*We shortened the paragraph, but kept the information on the tropospheric mean value valid for our measurements as well as the information on the annual increase and the stratospheric sink, since both are relevant for our study.*

Nitrous oxide ($N_2O$) has a lifetime of 123 years (Ko et al., 2013) and is released at the surface with no chemical sources in the atmosphere (Dils et al., 2006). As a results $N_2O$ has a near constant tropospheric value of 329.3 $ppb_v$ (winter 2015/2016 according to NOAA) that makes stratospheric influence identifiable (Müller et al., 2015). The mean tropospheric increase was found to be 0.78 $ppb_V$ (Hartmann et al.2013)

The main sink reactions of $N_2O$ are due to photolysis in the UV-band (190 nm  220 nm) and the reaction with $O(^1D)$ which only occurs within the upper stratosphere (Ko et al., 2013). Thus, $N_2O$ above the tropopause shows a weak negative vertical gradient which maximizes during winter and spring due to the diabatic downwelling by the Brewer-Dobson circulation. […]

**Paragraph beginning p. 9, line 32. A clearer way to say this: "Figures 4a and 4b show N2O values between 276-325 $ppb_V$ were measured during Phase 1 and values below 200 $ppb_V$ were measured during Phase 2 above 400 K. Figure 4c shows an overall decrease in N2O in the polar LS due to diabatic descent during winter, consistent with mean age changes (Fig. 3c)."**

*Sentence changed according to suggestion*

**Section 4.1.3 CO, first paragraph. This has way too much detail. Delete the 2nd sentence. 3rd sentence: "Due to the high variability of anthropogenic surface emissions, CO mixing ratios in the northern hemisphere vary from 70-200 ppbs [prinn] and the CO lifetime is on the order of weeks." 4th sentence ok. 5th & 6th sentences: "The main sink is oxidation by OH. The CO lifetime during polar night is a few months." Don't need reaction (1).**

*Changed according to suggestion*

**The phrase 'equilibrium value' is not right – this is not about chemical equilibria. I believe you are trying to describe a minimum stratosphere CO that comes from being**

**far above tropospheric CO sources but well below where high mesospheric CO can influence the lower stratosphere. Please delete usage of equilibrium CO (happens a lot on p. 13). Also, 'stratospheric background value' used on the next page and elsewhere is a vague expression. Can you say what you mean by this?**

*With equilibrium value we refer to the equilibrium between CO production from Methane oxidation ($CH_4$+OH is the rate limiting step) and the degradation of CO via OH. In the lower stratosphere there are two important reactions which determine the abundance of CO. The first one is the production of CO via methane oxidation and OH, the second one is the sink reaction of CO with OH to $CO_2$, which is much faster than the production from methane. Since both reactions are driven by OH, the CO concentration in the lower stratosphere depends on the available OH and methane. In the absence of transport from the troposphere and subsequent mixing CO will be degraded rapidly, but not to zero since the CO production from methane, which has a long lifetime in the stratosphere of (190 ± 50) years (Brown et al., 2013) acts as a source as long as methane and OH are available. A steady state equilibrium is the result according to [CO]_eq = k1/k2 [$CH_4$] after solving d[CO]/dt = k1[$CH_4$][OH] – k2[CO][OH] = 0. Note that the value of CO_eq is independent from OH , since [OH] cancels out. Therefore in the long-term limit CO-production partly compensates CO-destruction leading to a CO value of 5-15 ppb$_V$ depending on the integrated temperature history of the air masses, since the reaction rates depend on $k_i$. We termed the resulting CO value equilibrium value to emphasize, that we refer to a chemically driven equilibrium value. As long as methane is present in the stratosphere, CO can be produced.*
*Looking at tracer tracer correlations, the CO-equilibrium can be seen in vertical branches of correlations, when using CO as x-axis (see e.g. Fig. 9). This vertical branch indicates vanishing CO variability and has been observed e.g by Flocke et al., 1999, Herman et al. 1999 and Marcy et al., 1999.*
*Therefore we kept the term CO-equilibrium value.*

**p. 10, line 15. All you need to say is "The reaction of CH4 with Cl is an insignificant source of CO in the lower stratosphere [Flocke]."**

*Changed to the suggestion*

**p. 11, line 6. Figure 7 is mentioned before Figure 6. Reorder.**

*In this case both figures are relevant, we changed the reference to Figures 6 and 7.*

**p. 11, line 10. The CO change during winter at 340 K is not so consistent with N2O. The CO changes small and are both positive and negative in this region. The results are mixed, not clear.**
**The sentence refered to Fig. 5 a)and 5 b). It is correct that the differences in Fig. 10c)show changes in both directions at lower isentropes**

*At lower isentropes the tropospheric variability starts to affect the overall distribution. Since CO has a lifetime on the order of months part of the tropospheric CO variability affects the LMS beyond the ExTL due to the lifetime of CO. As described in the shortened $N_2O$ paragraph, $N_2O$ does not show the same tropospheric variability as CO and thus the $N_2O$ change between both phases (Fig. 4c) shows a much more homogeneous distribution at low isentropes.*
*We changed the sentence:*

Hence the overall distribution of carbon monoxide in the UTLS during the individual phases (Fig. 5ab) seems to be consistent to $N_2O$ and mean age obtained from $SF_6$ measurements, despite its much shorter lifetime compared to the other species.

**p. 11, line 15. Change 'rise the hypothesis' to 'suggest'**

*Changed*

**p. 11, line 17. Since there are no CO surface observations used, you can't really say anything about the strength of tropospheric source emissions.**

*Based on our observations, it is correct, that we can't directly account for surface emissions. We can however take our observations at low equivalent latitudes. We also included MLS data in the later discussion.*
*We changed the statement*

[...] from the tropical lower stratosphere over the course of winter without an increase of the upper tropospheric mixing ratios, which are affected by the surface emissions.

**p. 12, lines 2-5. I don't agree with the statement that the 'lowest values' are found in the 'furthest regions from the troposphere'. The difference plots (3c and 4c) show a range of differences (even a few increases), so if you were to quantify this (rather than showing color blocks) it's not obvious that you statement would be true. Can you be more quantitative, or at least modify the words here?**

*Exactly for this reason we want to keep the overall tracer distributions as a function of latitude and potential temperature since the lowest values are not on display in panels 3c) and 4c), which show differences. They are shown in Figure 3 b and 4 b) at Θ = 380 K. We also sharpened the text, since the statement about absolute values and changes were not clearly separated.*

We found a decrease of the long lived species $SF_6$ and $N_2O$ with their lowest values far above the local troposphere in late winter […]

**p. 12, line 4. 'contradicting' should be replaced, perhaps with 'unexpected'.**

*Changed*

**p. 12, lines 8-16. Here – and in general – there are too many sentences telling the reader what you are going to write about (or discuss later) rather than just writing about the subject. CO's use as a tracer has already been mentioned (Fig. 5) and the insignificant source form CH4 + CO has already been noted. Delete.**

*We followed the suggestion and shortened the paragraph:*

In the following we will discuss this hypothesis and also other potential sources for the additional CO mixing ratios. To identify mixing processes across the tropopause CO-$O_3$ correlations have been widely used (Fischer et al., 2000; Zahn et al., 2000; Hoor et al., 2002; Pan et al., 2004; Müller et al., 2016). Since ozone is affected by chemical processes particularly in the vortex region we use $N_2O$ as stratospheric tracer instead of ozone.

**p. 13. Line 1-2. Use active voice as much as possible, e.g., 'tropospheric data have high N2O [how high?] and are accompanied by high CO while stratospheric data have N2O < 328 ppb$_V$'.**

*Changed*

**p. 13, line 7. N2O is 'longer-lived' not 'chemical inert'.**

*We changed the phrase to:*

**p. 13, line 13. Regarding the line showing meso CO on Fig. 6d, it should point to a value of 0 N2O because there is essentially no N2O in the mesosphere.**

*Changed*

**p. 13, lines 23-28. This result is not so mysterious and it would not seem so if this were discussed in terms of the species' gradients. The changes observed during winter come from 2 processes: descent and mixing. The change in each tracer depends on the tracer's vertical and horizontal gradients as well as the balance between vertical and horizontal motions. This is a more physically meaningful way to explain the observations.**

*This is exactly true, that descent and mixing act on the tracer and their gradients. The statement is not in contradiction to our analysis. Differing from the classical Plumb and Ko (1994) regime using long-lived tracers the use of two tracers of very different lifetime (like CO and $N_2O$) introduces some additional information to help to distinguish between mixing from the tropopause region and descent of aged air (but of course does not obey slope-equilibrium any more).*
*Indeed, the CO gradient on $N_2O$ isopleths is used to unmask the different contributions of increased contribution of young air during our measurements. The quantification and relation to the age spectra later allow to further quantify these contributions.*

**p. 13, line 31. Try instead 'The region measured during both phases is 340-380K.'**
*Changed*

**p. 13 line 34-35. Why do you assume that the increased tropospheric fraction comes from the TTL? The POLSTRACC data didn't sample air south of 40N, so how can you be so certain of its origin? Isentropic mixing across the LMS tropopause (320-360K range) between the midlatitude UT and the polar LS would have the same effect, right? Without observations or a trajectory analysis, there isn't convincing evidence presented that the TTL is the source.**

*We checked this and added a plot which shows that the trajectories lead to the tropical tropopause region, particularly for the data above PV = 7 PVU. We used different tropopause definitions and identified the tropics and subtropics above $\Theta$ = 370 K as source regions for the trajectories.*

[Figure]

*Fig. R1 (a): Probability density function against minimum trajectory latitude*

[Figure]

*Fig. R1 (b): Probability density function for phase 1 (blue) and phase 2(orange). Left: less than 50 days in the stratosphere, right: at least 50 days in the stratosphere*

[Figure]

*Fig. R1 (c): Potential temperature of TST against latitude of TST for phase 1*

[Figure]

*Fig. R1 (d): Potential temperature of TST against latitude of TST for phase 2.*

*The trajectory analysis indicates that the fraction of air masses from the TTL increases for phase 2. This is particularly true for those air masses originating above Θ = 370 K.*
*1. Panel (a): Relative distribution of minimum latitude of backward trajectories, which undergo a TST within the last 50 days. The contribution of trajectories originating from the TTL region is larger in phase 2 compared to phase 1.*
*2. Panel (b): The relative amount of trajectories which stayed less than 50 days in the stratosphere is in phase 2 greater than in phase 1.*
*3. Panel (c) and (d): Potential temperature and location of TST points of trajectories indicate that most trajectories in phase 2 have TST in the region of the TTL and the tropical tropopause, whereas the TST location in phase 1 extends to lower isentropes.*

**p. 16. Equations 2, 3, and 4 use mixing rations and fractions. I don't see any use of mass here. Why then is this described as mass balance?**

*The air mass density cancels out for the final equation 4 since the ratio of mixing ratios is calculated.*

**p. 16, line 15. Why are you citing 10-20 year old studies for the value of CO at the tropical tropopause? You could be using something far more precise and current by looking at MLS CO measurements (2004-present, as low as ~ 300 hPa). CO at the tropical tropopause has considerable seasonal variation due to seasonal biomass burning influence. MLS shows CO above 100 ppb$_V$ in the TTL much higher than what is used in your study – see Huang et al. [ACP, 2016]. This assumed TTL CO value impacts your results. This section needs to be revised based on recent measurements of TTL**

*Since we use in-situ data we want to compare our measurements to in-situ measurements of other data sets, regions and years despite the disadvantage of the small data coverage during an aircraft based campaign. We appreciate the availability of satellite observations which provide a global view. For quantitative comparisons of aircraft observations with satellite data averaging kernels have to be applied, which is difficult in our case since most of the data were measured along horizontal flight tracks. In addition we are detecting rather small changes of CO on the order of 3 ppb$_V$.*

*As stated in Huang et al. (ACP 2016), the estimated single measurement precision for CO is 19 ppb$_V$ and the systematic uncertainty is on the order of +/- 30 % for CO (MLS V4.2) based on Livesey et al., (2015).*

*Zonal and seasonal means as shown in Huang et al., (2016) however provide the possibility to compare our data qualitatively to the observations by MLS.*

*As can be seen in Huang et al., (2016) Fig. 12 and 13 CO climatologies show maximum values of 110 ppb$_V$ at 215 hPa and 80 ppb$_V$ at 100 hPa with higher peak values in certain years or regions. To our knowledge there is no airborne data set, which shows such high mean CO values in the tropics besides individual plume encounters.*

*Therefore we performed our calculations for CO values of 60, 70 and 80 ppb$_V$ according to the climatological regional profiles (Fig.14 in Huang et al., (2016)) at 100 hPa. The estimates of our calculated fractions are changing by 2.2% and are shown below. Since the denominator of our equation (3) depends linearly on the difference between CO (tropical) and the observed stratospheric background an increase of tropical CO from 60 ppb$_V$ to 80 ppb$_V$ reduces our fraction by 32 %.*

*We accounted for this by indicating the range for different estimates of the CO-tropical and modified the text.*

[Figure]

*Fig. R2: N₂O against the difference of the trop. Fraction from phase 1 to phase 2 for different tropical CO entry values in the TTL*

The CO increase over the course of winter corresponds to an increase by f_trop of (6.8 +/- 3.7)% between 313 ppb$_V$ and 273 ppb$_V$ N₂O by assuming 60 ppb$_V$ of CO at the tropical tropopause as provided by in-situ aircraft data from Herman (1999) and Marcy et al. (2007). Using CO_trop = 80 ppb$_V$ as indicated by MLS at 100 hPa one obtains 32% lower values for f_trop, which is still a significant increase of tropospheric air masses

**p. 16, lines 28-29. Should be 'longer transit times' and 'mixed with young tropospheric air with shorter transit times ... '**

*Changed to the suggestion*

**p. 17, line 17. Try 'Figure 10b shows there is ....' 0.3% is a very small change. What is the uncertainty on this number? Is 0.3% statistically different from zero?**

*Changed to the suggestion.*

*We thank the reviewer for this point indicating a typo. The relevant unit in Fig. 10 is '% per month' and was not given in the text. The correct phrase now gives '…0.3 % per month…' as indicated by the vertical axis label in Fig.10. Note, that these 0.3 % per month are absolute values. The age spectra in Fig. 10 a) indicate the maximum difference of 0.3 % per month at e.g. 3.5 years transit time (panel (b)). The absolute value in panel (b) for phase 1 is 0.41 % per month and for phase 2 is 0.71 % per month. Therefore the relative increase in the age spectra*

*is 76 % which is significant.* The relative increase of the young fraction (less than six month) is 19 %.

**p. 18, line 2. Try '... the observed CO increase, indicating increased mixing with air..'**

*Changed to the suggestion*

**p. 18, line 7. This is really the lowermost not the lower stratosphere you are talking about.**

*Changed to:*

[…] the overall ageing in the lower and lowermost stratosphere over […]

**p. 18, lines 11-12. Try 'To further investigate the relationship of young versus aged air, we calculate the accumulated fraction ...'**

*Changed to the suggestion*

**p. 18. The paragraph beginning on line 15 should be part of the previous one.**

*Changed*

**p. 21, line 1-3. You could make a good point here that mean age is an incomplete or oversimplified way to characterize air. The age spectrum shows there changes in the contributions from older and younger age masses that determine the change in mean age.**

*We added a sentence to the manuscript:*

Therefore our results demonstrate, that the mean age is an incomplete descriptor when referring to chemical properties of air masses involving different chemical life times of species. Since the mean age is just a single number it might be insensitive to changes of the processes and time scales contributing to the mean, but affecting chemical properties and impact of the air parcel. Therefore it is important to account for the full spectral shape when referring to chemical properties of an air mass rather than only the mean age.

**p. 21, lines 5-10. You can cite Rinsland 1999 and Rosenfield 1994 that mesospheric CO just doesn't go this low into the stratosphere, and then you are done with this issue. The CH4+CL sources of CO has already been labeled insignificant. There is no need to talk about these insignificant sources again. This means that all of Section 6 (Discussion) before about p. 23 can be deleted.**

*We shortened the discussion. We kept the individual points in the manuscript, since the observed change of CO relative to $N_2O$ is small and we have to exclude potential sources of CO. Further the reaction $CH_4+Cl$ is in principle able to produce the respective amount of CO according to CLaMS. The additional CO is however reacting via OH leading to a zero net increase. We only showed the net effect in the manuscript to keep it short.*

Carbon monoxide is produced in the mesosphere due to the photo-dissociation of carbon dioxide. Therefore the composition of mesospheric air masses is clearly distinct from air mass composition of the stratosphere. Rinsland et al. (e.g. 1999) found increased CO mixing ratios up to 90 $ppb_V$ at altitudes around 25 km or $\Theta = 630K - 670$ K and Engel et al. (2006b) found CO values of 600 $ppb_V$ at an altitude of 32 km. Both studies show very low $N_2O$ mixing ratios (< 50

ppb$_V$). Although the authors found layers of mesospheric air descending down to 22 km, this is not evident for the Arctic winter 2015/16 and lowest N$_2$O mixing ratios are found to be in the order of 200 ppb$_V$.

This is reflected in the MLS observations that determine the CLaMS upper boundary at $\Theta$ = 900 K potential temperature (Fig 12.). The simulation indicates the expected downward transport of mesospheric influenced air, but down to $\Theta$ = 600 K at the end of March 2016 in agreement with our observations which minimize at the highest flight levels and equivalent latitudes. Furthermore, an additional influence of descended mesospheric air into the lower stratosphere would lead to mixing lines very strongly differing from the observed relationship (see Fig. 6), which is not observed in agreement with the CLaMS N$_2$O-CO scatter plot (Fig. 9).

In general, another important source of carbon monoxide in the atmosphere is the reaction of methane with reactive chlorine, which is not significant in the lower stratosphere (Flocke et al., 1999). However, air masses enriched in reactive chlorine could have been transported downwards, providing potentially reactants for the chemical production of CO. Therefore, we simulated the CO yield from the reactions of CH$_4$ with chlorine, OH and O($^1$D) using CLaMS simulations in the box model mode. A large number of air parcel backward trajectories starting on 15 March from locations within the vortex core (equivalent latitude > 65 °N; potential temperature between $\Theta$ = 350 K and $\Theta$ = 500 K) ending on 15 January and chemical composition changes were calculated using the CLaMS chemistry module running forward in time for a subset of the trajectories with equivalent latitudes > 50 °N on 15 January (21480 trajectories).

Figure 13 shows the statistical evaluation of the net CO change due to chemistry over the period as function of potential temperature on 15 March. The blue line represents the statistical mean and the dashed lines the 1-sigma standard deviation. The mean overall change is even negative over the entire profile, which is due to the oxidation of the produced CO by the reaction with OH. Therefore we conclude that the observed increase of CO in phase 2 is not due to the additional chemical source reaction.

Additionally the age spectrum calculations of the CLaMS model provide mass fractions of air masses regarding their stratospheric residence time. As is evident from Fig. 9 there is a significant increase of air masses younger than six months at typical mean ages for lower stratospheric air masses and mesospheric influence on the basis of our analysis is highly unlikely.

**p. 23, 10. Here's a suggestion for Fig. 14. Since you say that the SH results should not be compared to the NH, then don't show them. Figure 14 would be more readable if you just showed the regions of interest. Try making a 4-panel figure (Jan and March), 0-90N or 30-90N only.**

*We changed the Figure according to suggestion*

**p. 25, line 5. What are the significant figures here? Is 0.29 really different from 0.27, or are they both 0.3?**

*The increase by 0.29 years of SF$_6$ is deduced from the measured SF$_6$ distribution in Fig. 3c) and the increase by 0.27 years is calculated from CLaMS mean ages along the flight track in Fig. 10 a) which is now changed in the manuscript. (see also reply to reviewer 1):*
*Note that the increase of 0.29 years is only valid for the overlapping distribution of phase 1 and phase 2 and also the CLaMS simulations indicate an increase of the mean age of the same magnitude, which is consistent.*

*The Figure below further illustrates the change of the mean age distribution towards higher mean ages based on the SF$_6$ distributions for the two phases. The increase of the mean age over all observed data of the distribution is 0.79 years.*

[Figure]

*Fig. R3: Box-Whisker plot of Mean Age (SF₆) for phase 1 and phase 2*

**p. 25, line 18. The conclusions should be about the lowermost, not lower stratosphere as all the results are below 380K**

*We changed our manuscript accordingly.*

*References:*

*M. P. Baldwin, L. J. Gray, T. J. Dunkerton, K. Hamilton, P. H. Haynes, W. J. Randel, J. R. Holton, M. J. Alexander, I. Hirota, T. Horinouchi, D. B. A. Jones, J. S. Kinnersley, C. Marquardt, K. Sato, and M. Takahashi, "The quasi-biennial oscillation," Rev. Geophys., vol. 39, no. 2, pp. 179–229, May 2001.*

---

## Author Comment (AC1)

We thank Referee #1 for her/his comments and careful reading and changed the manuscript according to her/his suggestions. Our response is formatted as follows:

**Referee's comments**

*Author's reply*

Changes to the manuscript

**The paper uses aircraft tracer measurements from the POLSTRACC campaign combined with simulations with the CLaMS model to derive conclusions on the characteristics of transport in the high-latitude boreal lower stratosphere for the winter of 2015-16. The results show an increase of the mean age in the region from January to March 2016, which seems at odds with the increase in CO concentrations. The authors argue that this is due to a change in the age spectrum, which exhibits an increase in both old and young air by the end of the winter.**

**The article is well written and presents an interesting analysis motivated by in situ observations and nicely complemented with modeling tools. I recommend publication in ACP after the following few minor comments and technical corrections are addressed.**

> **The main point that should be addressed regards the high values of CO observed during phase 1, seen in Fig. 5a at about 70 degrees and 330-340K, and in Fig. 7 (CO values above 45 ppb$_v$). Although these values are not the main focus of the paper, they stand out, and there are a few parts in the paper where I miss some explanation of their origin. For instance, in the description on Page 13 Lines 25-29 it is mentioned that the 'direct tropospheric impact was greater in phase 1 than in phase 2' referring to these points. What do you mean by 'direct tropospheric impact'? Is this transport across the ExTL or did the high CO values originate in the TTL?**

> *The expression 'direct tropospheric impact' should indicate, that the decrease of CO relative to $N_2O$ occurs at the highest stratospheric values of $N_2O$ (i.e. at most tropospheric influenced air masses), where air parcels which have been recently transported into the lowermost stratosphere have in general the shortest stratospheric residence time.*

> *It is not possible to derive from Figure 7 the information whether transport out of the TTL region occurred in this specific case or from the ExTL. The values you are referring to were encountered during one specific flight (PGS 09) on 22.01.2016. The flight track crossed a filament of air with relatively high values of tropospheric trace gases and a high tropopause with a sharp PV gradient. However, since small scale processes like gravity waves, occurrence of turbulence in regions of strong wind shear at the jet or diabatic heating violate adiabatic PV conservation this may lead to a mismatch of analyzed PV fields and tracer occurrence, which could also have caused the anomalously high CO values in this case.*

> *Since we analyzed our dataset only for PV > 7 PVU, we expect that the overall impact of the ExTL on our analysis is small.*

**Also in Section 5.2, you could look separately at the age of air spectrum for those air masses, instead of showing the results for all measurement points in phase 1 together. Does that help in interpreting the origin? Finally, some measurements in phase 1 were taken at lower latitudes (over Italy) compared to the rest of the campaign. Does that latitude difference have an impact on the CO values?**

*The measurements over Italy do not affect our analysis of the observed CO increase relative to $N_2O$. Due to technical problems, we were not able to obtain $N_2O$ measurements during this flight, so these data points do not appear in the CO-$N_2O$ correlation and our analysis. Furthermore, these data points would be excluded by applying the 7 PVU criterion to our data as described in the manuscript.*

*The number of data points with high CO between 330 K and 340 K in this region is 297, compared to 5518 data points for the whole distribution of phase 1, which just makes a fraction of 5.3%. Since the data were observed in a region of strong PV gradients as described above and the main focus of the paper is on the region above $\Theta = 340$ K, we did not analyze these age spectra separately.*

We changed Fig. 2 of the manuscript. We now distinguish between parts of the flight track below PV = 7 PVU and above. We further removed the flight over Italy since the $N_2O$ data are missing and do not contribute to our measurements.

**P1 L22: 'diabatic descent [. . .] adds to the diabatic downwelling of the Brewer-Dobson circulation'. It seems to me you are referring to the same thing twice?**

*We wanted to refer to the two main processes which lead to diabatic descent during the polar night over the poles, namely the absence of radiation and associated diabatic cooling and the wave driven descent. We changed the section to:*

Diabatic descent in the polar stratosphere, which is strongest inside the polar vortex results as part of the Brewer-Dobson circulation (Brewer, 1949; Dobson, 1956) in mid and high latitudes as response to the breaking of planetary and gravity waves (Haynes, 1991; Plumb, 2002; Butchart, 2014) in the upper stratosphere and mesosphere.

**P2 L29: tropical pipe**

*Sentence changed to:*

The region between $\Theta = 380$ K and the bottom of the tropical pipe around $\Theta = 450$ K (Palazzi, 2011) is a key region for the transition between these transport regimes.

**P3 L23: The McPhaden reference is not about the 2015 ENSO event. A better option could be perhaps L'Heureux et al. (2017).**

*References changed to Chen et al. (2016) and L'Heureux et al. (2017)*

**P3 L24: The impact of the 2015-16 ENSO event on the polar vortex has been analyzed by Palmeiro et al. (2017).**

*Manuscript changed to:*

A direct influence on the polar vortex is still under debate and according to Matthias (2016) this strong El-Niño is suggested to account for a weakening of the polar vortex, while Palmeiro (2017) found a connection of this ENSO event to the strong polar vortex and the early MFW.

**Only flights that were used for the analysis are shown.**

*Caption changed to your suggestion.*

**P8 L1: take → taken**

*Changed*

**P8 L20: remove 'respective'**

*Changed*

**P8 L27: remove respectively?**

*Changed*

**P9 L7-8: is this a hypothesis or do you have an argument to support this statement?**

*This is a hypothesis based on the assumption that a change in the lifetime of $SF_6$ would lead to an equal change in the absolute values of mean age for both phases of the campaign. This assumption is also supported by the fact that we do not see any indication of an influence of mesospheric air on our observations. Since the discussion in our study is based on relative changes of mean age between phase 1 and phase 2 it is unlikely that the mesospheric loss of $SF_6$ affects the differential analysis of the calculated mean age.*

**P9 L14: Are the physical altitude ranges the same for both phases?**

*During both phases of the campaign the flight profiles were very similar and nearly every flight reached FL450 to FL480 (pressure altitude ranges to 45000 ft and 48000 ft, respectively).*

**P9 L18: The mean increase of 0.29 is just below the precision of the mean age estimate from SF6. Do you still consider it a robust change?**

*We consider the change as significant. Even if the observed change in the estimated mean age from $SF_6$ is just below the precision for each individual data point, one obtains a significant change in the mean binned mixing ratios of $SF_6$. Note that the increase of 0.29 years is only valid for the overlapping distribution of phase 1 and phase 2 (Fig. 3c)) and also the CLaMS simulations indicate an increase of the mean age of the same*

*magnitude, which is consistent.*

*The Fig. R1 below further illustrates the change of the mean age distribution towards higher mean ages based on the $SF_6$ distributions for the two phases. The increase of the mean age over all observed data of the distribution is 0.79 years.*

[Figure]

*Fig. R1: Box-Whisker plot of Mean Age ($SF_6$) for phase 1 and phase 2.*

**P9 L27: "with an variability" → an interannual variability?**

*Manuscript changed to:*

a meridional variability

**P11 L2: chapter → section**

*Phrase „chapter" changed to "section".*

These potential influences are discussed in section 6.

**P12 L4: [...] general picture of enhanced downwelling of the Brewer-Dobson circulation [...]**

*Manuscript changed according to the suggestion*

[…] which fits well in the general picture of enhanced downwelling of the Brewer-Dobson circulation in late winter/spring.

**P12 L9: despite the**

*Changed*

**P13 L4-13: Could you refer to the individual panels of Fig. 6 as you describe the figure?**

*Changed*

**P13L21: anCO→aCO**

*Changed*

**P13 L25-29: This description is unclear. What you mean by 'shows higher CO relative to N2O?'. Perhaps it would make it easier to follow if you referred to the isentropic levels approximately corresponding to the N2O values when you describe Fig. 7 (it is hard for the reader to combine mentally Figs. 4, 5 and 7).**

*This sentence shall highlight the main result of this figure and refers to Fig.7. Higher CO values relative to $N_2O$ are evident between $N_2O = 275\ ppb_V$ to $320\ ppb_V$. Compared to Fig. 6c), this is an indication that this change of the correlation can only be due to a change of the effectiveness of mixing. At this point we leave the geometric (or isentropic) coordinates since the tracer coordinate $N_2O$ in Figure 7 serves as natural tropopause following coordinate.*

*If one would try to deduce the results from isentropic coordinates one could not differentiate between mixing and transport processes. Since both tracers will undergo the same transport and mixing processes these processes are accounted for in tracer tracer correlations. Relative changes of two tracers of very different lifetime like CO and $N_2O$ therefore indicate changes of either sources or sinks or the transport efficiency.*

**P17 L4-5: would it be more accurate to refer to these figures as 'scatter plots' rather than 'correlations'? Also on Fig. 9 caption.**

*Changed to the suggestion*

**P17 L18: remove 'which is'**

*Changed*

**P18 L8: Green's function**

*Changed*

**P21 L1: what do you mean by 'mass balance systems of transport pathways'?**

*This refers to equation (4) and is changed to:*

[...] the mass balance equation

The analysis of the CO-N₂O correlation and the mass balance equation as well as the model simulations consistently point towards …

**P22 L27: Eventhough → Although**

*Changed*

**P24 L8-10: This sentence is unclear. Do you mean that the high fraction of young air reaches higher latitudes in 2015/16 as compared to the climatology? If so, what is the variability (e.g. standard deviation) around the climatology? Is this winter statistically different from the climatology?**

*The below graph R2 shows line plots of the relative difference of air masses with transit times smaller than six months (MF06) from March 2016 to March of the climatology (thick line) at 350 K and 400 K.*
*At latitudes northwards 60° there are up to 10% more MF06 air masses as compared to the climatology, which also supports our hypothesis.*

[Figure]

*Fig. R2: Line plot of difference (thick line) between March 2016 and March of the climatology. Dashed lines denote the standard deviation.*